

# Estimation of path attenuation and site characteristics in the north-west Himalaya and its adjoining area using generalized inversion method

Harinarayan, Nelliparambil Hareeshkumar[1], Abhishek Kumar[*2]

[1, 2] Department of Civil Engineering, Indian Institute of Technology Guwahati, Assam, India.

Corresponding to: Kumar Abhishek (abhitoaashu@gmail.com/ abhiak@iitg.ernet.in)



**Abstract.** Present work focuses on the determination of path attenuation as well as site characteristics of
PESMOS managed recording stations, located in the north-west Himalaya and its adjoining region, using two-
step generalized inversion technique. In the first step of inversion, non-parametric attenuation curves are
developed. Presence of a kink is observed at around 105km hypocentral distance while correlating the path
attenuation with the hypocentral distance indicating the presence of Moho discontinuity in the region. Further,
$Q_s = 105\, f^{0.94}$ as S wave quality factor within 105km, is obtained indicating that the region is possibly
heterogeneous as well as seismically active. In the second step of inversion, site amplification curves are
developed separately from the attenuation corrected data for horizontal and vertical components of the
accelerogram. Further, site amplification spectra is computed as the ratio of the obtained horizontal and vertical
components to determine the amplification function and predominant frequency for each of the PESMOS
managed recording stations, exist within the study area. The predominant frequency estimated by generalized
inversion method are in good agreement with those obtained using horizontal to vertical spectral ratio of the S
wave portion of the accelerogram. Maps showing spatial distribution of predominant frequencies and
amplification functions across the study region are also developed based on the present work.
Keywords: *PESMOS recording stations, Northwest Himalaya, path attenuation, site characteristics,*
*Generalized Inversion, HVSR*



## 1 Introduction

The Himalayan arc extends approximately 2500km starting from Kashmir in the northwest to Arunachal Pradesh in the northeast and is considered as one of the most seismically active regions in the world. This region has experienced several destructive earthquakes (EQs) including 4 great EQs (1897 Shillong EQ, 1905 Kangra EQ, 1950 Assam EQ and 1934 Bihar-Nepal EQ) in the last 120 years. Based on the seismic activity, the entire Himalayan belt can be subdivided into three distinct segments namely the western, the central and the eastern segments (Philip, 2014). The region of the north-west Himalaya and its foothills within India encompassing the states of Punjab, Uttarakhand, Delhi, Haryana and Himachal Pradesh come under seismic zone IV and V as per IS 1893: 2002 indicating high to very high seismicity. Therefore, the necessity of precise seismic hazard studies of this region has become an issue of great importance.

The intensity of ground shaking during an EQ at a particular site is a function of source, path and site parameters. Source parameters includes magnitude, fault mechanism, stress drop and rupture process. On the other hand, path parameters include geometric attenuation and loss of seismic energy due to the anelasticity of the earth and scattering of elastic waves in heterogeneous media. Similarly, site characteristics include modification of amplitude, frequency content and duration of the incoming seismic wave by subsurface medium as reached to the surface. Determinations of aforementioned EQ parameters are important for development of region specific GMPES which can be used for region/ site specific seismic hazard assessment. Above parameters are commonly estimated using EQ records and some spectral modelling or inversion approach like generalized inversion method (Andrews 1986; Castro et al., 1990; Oth et al., 2009 etc.).

In the present study, EQ records from the region of the north-west Himalaya and its foothills, obtained from PESMOS database are analysed for estimating path attenuation and site characteristics separately, using a two-step generalized inversion of the S-wave amplitude spectra (hereafter referred to as GINV). In the first step, attenuation curves are developed using a non-parametric inversion approach similar to Castro et al., (1990) and Oth et al., (2008). In the conventional generalized inversion method (Andrews, 1986; Hartzell, 1992), site and source terms are estimated simultaneously in the second step of inversion, by inverting the S-wave (or Coda wave) spectra corrected for the path parameter. This method requires a reference site (whose site amplification is known) in-order to remove the trade-off between the source and site parameters (Andrews 1986). In case of PESMOS database, however, due to the lack of detailed knowledge about the geology beneath the recording stations, identifying a reference site is not possible. In the absence of a reference site, only the site parameter is evaluated in the second step of inversion, using a non-reference generalized inversion approach (similar to the work by Joshi et al., 2010; Harinarayan and Kumar 2017). The obtained site terms are compared with the horizontal and vertical ratios (HVSR) calculated from the same S-wave window as used in the GINV above.

This study is one of the first attempts in the area to systematically evaluate path and site parameters using a larger database. Available sudies on attenuation characteristics of north-west Himalaya are in fact based on considering few EQ records from limited recording stations. These include; Joshi (2006) estimated frequency independent S wave quality factor ($Q_s$) for the Garhwal Himalayas using 1991 Uttarkashi EQ and 1999 Chamoli EQ ground motions from 8 recording stations. In another study, Singh et al., (2012) estimated frequency depended $Q_s$ for the Kumaun Himalays using 23 EQ events, from 9 recording stations applying the extended coda-normalization method. Similarly, Negi et al., (2015) and Tripathi et al., (2015) estimated $Q_s$ for the



Garhwal Himalayas. The aforementioned studies did not highlight the attenuation characteristics of the entire
north-west Himalaya region. In addition, similar to path attenuation studies, very few literatures on the
determination of site characteristics from EQ records have been reported for this region.  These include work
Nath et al., (2002) which computed site terms using the aftershocks of the 1999 Chamoli EQ from 5 recording
stations located in the Uttarakhand region. Similarly, by Sharma et al., (2014) using EQ records in context of
generalized inversion and horizontal to vertical spectral ratio to estimate site parameters for the Garhwal region
of Uttarakhand. In another work, Harinarayan and Kumar (2017) reported a comparative study on site
characteristics computed using EQ records from Tarai region of Uttarakhand using multiple analytical
approaches. In another recent work, Harinarayan and Kumar (2018) computed site parameters in terms of
predominant frequency ($f_{peak}$) alone, utilizing the EQ records from the north-west Himalaya and nearby regions
using spectral ratio method. Site characteristics in terms of amplification function ($A_{peak}$) for the recording
stations have been missed in Harinarayan and Kumar (2018). In the present study, site characteristics are
determined in terms of $f_{peak}$ and $A_{peak}$. Further, maps showing spatial distribution of $f_{peak}$ and $A_{peak}$ are separately
developed for regions of Delhi, Uttarakhand, Punjab and Himachal Pradesh in this work. Such maps can be of
utmost importance for seismic site classification, ground response analyses and microzonation studies in the
future works.
**2 Database**
The input data used in this study consists of three components accelerograms obtained from PESMOS database
available at http://www.pesmos.in/. The instrumentation used for recording EQs consists of internal AC-63
GeoSIG triaxial force balanced accelerometers and GSR-18 GeoSIG 18 bit digitizers with external GPS (Kumar
et al. 2012). Ground motion recordings are done in trigger mode during each EQ with a sampling rate of 200 per
second.

In the present analysis, ground motions records corresponding to EQs happened between 2004 and

2017 are used. For estimating site characteristics, 341 records from 86 EQs, with magnitudes ranging from
Mw=2.3 to Mw=5.8, having focal depths ranging from 2 to 80km are used. Further, these records are
corresponding to 101 recording stations, located in the hypocentral distance ranging from 10 to 85km.
Coordinates of each of the recording station, used in this work are listed in Table 1, columns 2 and 3. Further,
details of EQs used for estimating site characteristics are summarized in Table 2.

For estimating path attenuation however, only those EQs, which are recorded at atleast two recording

stations in which at least one recording station should be located within hypocentral distance equal to or less
than the reference distance (reference distance is discussed  under section 'Spectral attenuation with distance)
are considered. Thus, not all recording stations whose records are used for computing site characteristics satisfy
the above reference distance criteria, are considered in path attenuation determination. Out of 341 EQ records
used for estimating site parameters, only 207 EQ records satisfies the reference distance criteria. Satisfying the
reference distance criteria, the database for estimating path attenuation consists of 207 records from 32 EQs,
with magnitude ranging from Mw= 3.1 to Mw=5.5, with focal depths ranging from 3 to 20km, recorded at 69
recording stations and within hypocentral distance ranging from 9 to 200km. Table 3 summarizes the details of



the dataset used for estimating path attenuation. In addition, Figure 1 shows the source-to-recording station path
coverage of all the data set used in the present study.

**2.1 Data processing**

All the EQ records collected above, are corrected for baseline correction following a 5% cosine taper and a
band-pass filtering, between the frequency range of 0.25Hz and 20Hz, using a Butterworth filter. Further, time
windows starting about 0.5s before the onset of the S wave and ending when 90% of the total seismic energy of
the EQ record is reached, are separated and tapered with a 5% cosine window (Ameri et al., 2011; Bindi et al.,
2009). Typical lengths of the time windows for the present analysis vary from 4 to 15s. Further, for some of the
records, where the window length obtained is longer than 15s, it is fixed to 15s in order to avoid a record having
too much of coda wave energy in the analysis time window (Oth et al., 2008). Later, based on the extracted
windows, the Fourier amplitude spectra is calculated for each EQ record, smoothened by applying the Konno
and Ohmachi (1999) algorithm, with the smoothing parameter "b" = 20.
For further analysis, path and site parameters are estimated using two separate inversion procedures using earlier
discussed EQ database, as discussed separately in the following sections.

**3 Path attenuation**

In the first step of inversion, spectral path attenuation curves are developed by eliminating the effect of site
parameter, thereby retaining only the source and path attenuation characteristics. All EQ records, irrespective of
whether located on soil or rock site can be utilized for inversion. Thus, present method is very much suitable for
PESMOS database where accurate site characteristics of recording stations are not available. The horizontal
portion of the accelerograms, obtained by the root mean square average of the east-west and north-south
components is considered for developing path attenuation curves. Detailed discussions on the method can be
found in following sub-section.

**3.1 Methodology**

Following Castro et al., (1990), observed spectral amplitude (acceleration) $U_{ij}(f, R_{ij})$, of EQ $j$, at recording
station $i$, and frequency $f$ can be modelled linearly as:
$$lnU_{ij}(f, R_{ij}) = lnM_i(f) + lnA(f, R_{ij}) \qquad (1)$$
Here, $M_i(f)$ is a scalar, which depends on the size of the EQ (one value for each EQ). Further, $A(f, R_{ij})$ is the
empirically determined attenuation function independent of the size of EQ, which incorporates both geometric
spreading and anelastic attenuation variation with the hypocentral distance. It has to be mentioned here that
$A(f, R_{ij})$ in Eq. (1) is not limited to a particular functional form, instead, is assumed to decay smoothly with
hypocentral distance ($R_{ij}$) and take the value of unity at a reference distance ($R_0$), i.e., $A(f, R_0) = 1$ (Castro et
al., 1990; 1996; 2003).
Model given by Eq. (1) does not contain any factor related to site effect. This site effect is absorbed in both
$A(f, R_{ij})$ and $M_i(f)$ and hence any rapid undulations in $A(f, R_{ij})$ are due to this absorbed site effects (Oth et
al., 2008). Two weighing factors, $\omega_1$ and $\omega_2$ are incorporated in the Eq. (1) following Castro et al., (1990). $\omega_2$ is



used to smoothen the attenuation term with distance curve by supressing the undulations and there by removing
any absorbed site effects from $A(f, R_{ij})$ and $\omega_2$ is used to impose $A(f, R_0) = 1$ constraint, as mentioned earlier.
The value of $\omega_1$ and $\omega_2$ is chosen reasonably such that the site effects are supressed and yet preserve the
variations of the attenuation characteristics with distance (Oth et al., 2008). In the matrix form, following the
notations of Menke (1989) and incorporating the weighting factors $\omega_1$ and $\omega_2$, Eq. (1) can be written in
accordance with Castro et al., (1990) as:

(A)                                            (X)             (b)

$$
\left[
\begin{array}{cccccc}
1 & 0 & 0 & . & \dots \\
0 & 1 & 0 & . & \dots \\
. & . & . & & \dots \\
1 & 0 & 0 & . & \dots \\
. & . & . & . & . \\
. & . & . & . & . \\
. & . & . & . & . \\
\omega_1 & 0 & 0 & . & \dots \\
-\omega_2/2 & \omega_2 & -\omega_2/2 & . & \dots \\
0 & -\omega_2/2 & \omega_2 & -\omega_2/2 & \dots \\
. & . & . & & \dots
\end{array}
\left|
\begin{array}{ccccc}
1 & 0 & 0 & . & \dots \\
1 & 0 & 0 & . & \dots \\
. & . & . & . & \dots \\
0 & 1 & 0 & . & \dots \\
0 & 1 & 0 & . & \dots \\
. & . & . & . & . \\
. & . & . & . & . \\
. & . & . & . & \dots \\
. & . & . & . & \dots \\
. & . & . & . & \dots \\
. & . & . & . & \dots
\end{array}
\right]
\right.
\left[
\begin{array}{c}
\ln A_1 \\
. \\
. \\
. \\
\ln A_{10} \\
\\
\ln M_1 \\
. \\
. \\
. \\
\ln M_N
\end{array}
\right]
=
\left[
\begin{array}{c}
\ln U_{11} \\
. \\
. \\
\ln U_{ij} \\
. \\
\\
0 \\
0 \\
0 \\
0 \\
.
\end{array}
\right]
$$

(2)

The hypocentral distances of the data set is discretized into number of bins of equal lengths and the value of
$A(f, R_{ij})$ is computed at each bin. The lengths of the bins are selected such that there is almost equal number of
data points in every bin. Further, $\ln A(f, R_{ij})$ versus hypocentral distance curves at each of the selected
frequencies are computed solving Eq. (2) in a least square sense, using singular value decomposition method
(Menke, 1989).
**3.2 Spectral attenuation with distance**
Figure 2 shows the number of EQ records for various hypocentral distance range considered. It can be observed
from Figure 2 that there are very few EQ records available beyond 115km. For this reason, EQ records with
hypocentral distance up to 115km are considered for the determination of path attenuation. The constraint
$A(f, R_0) = 1$ is applied at $R_0$=15km, irrespective of the frequency. The hypocentral distance range from 15 to
115km is divided into 10 bins, each bin having 10km width. Further, attenuation curves are computed for each
of the selected 17 frequencies from 1Hz to 15Hz (see Table 4, column 1). Variation of attenuation curves with
hypocentral distance, obtained in the present study, for the selected frequencies can be depicted in Figure 3.
Based on Figure 3, a general trend in which attenuation curves exhibit decay with distance up to 105km can be
observed, beyond which a kink is observed. The kink in the attenuation curves beyond 105km is very distinct
and clear at lower frequencies (<5.5 Hz). Bindi et al., (2004) and Oth et al., (2010) reported a similar trend in
the attenuation curves for the Umbria Marche and Japan regions respectively. Oth et al., (2010) attributed this
behaviour to the combined effect of reflected or refracted arrivals from the Moho in Japan. Presence of Moho in





the North-west Himalaya was reported by Saikia et al., (2016) based on Teleseismic receiver function analysis.
The above discussions suggest that attenuation curves obtained in this study at larger distances may be
influenced by reflected or refracted waves from the Moho. Observing the attenuation curves at different
frequencies in Figure 3 can conclude that at higher frequency, attenuation curves decay more rapidly than at
lower frequency. This observation is consistent with the findings by Castro et al., (2003) for the region of
Guadeloupe, France and Oth et al., (2011) for the region of Japan.
Further, for the kink observed at 105km, in case of lower frequencies, its sharpness reduces with increasing
frequency as can be observed in Figure 3. At frequencies greater than 10Hz, the kink at 105km smoothen and
the attenuation curves beyond 105km for higher frequencies becomes flat as observed in Figure 3. This change
in the character of the kink at higher frequency indicates that the arrival of waves from the Moho also gets
attenuated more at higher frequencies compared to lower frequencies.
**3.3 Quality factor estimation**
In order to estimate $Q_s$, inversion is repeated, however only considering records within hypocentral distance in
the range 15km to 105km, where a monotonic decrease in attenuation curves with distance is observed. The
attenuation curves are modelled in terms of geometric spreading $[G(f, R_{ij})]$ and quality factor $(Q)$ in accordance
with Castro et al., (1996) as;
$$A(f, R_{ij}) = G(f, R_{ij}) \left[ e^{\frac{-\pi \cdot f \cdot R_{ij}}{Q \cdot \beta}} \right] \tag{3}$$
Where, $f$ is the frequency and $\beta$ is the mean shear wave velocity in the crustal medium taken as 3.5km/s as per
Mukhopadhyay and Kayal, (2003). Further, $G(f, R_{ij})$ is considered as $1/R_{ij}$ in accordance with Banerjee and
Kumar (2015) for this region. For each frequency considered in this study (see Table 4), Eq. (3) is linearized by
taking logarithm and corrected for the effect of $G(f, R_{ij})$ as given in Eq. (4).
$$ln\, A(f, R_{ij}) - ln G(f, R_{ij}) = \frac{-\pi \cdot f}{Q \cdot \beta} R_{ij} \tag{4}$$
Ascribed to Castro et al., (2003), Eq. 4 is written in the form;
$$a(R) = -m\, R \tag{5}$$
Where $a(R)$ and $m$ are given as;
$$a(R) = lnA(f, R_{ij}) - ln\, G(f, R_{ij}) \tag{6}$$
$$m = \frac{-\pi \cdot f}{Q \cdot \beta} \tag{7}$$
Where, $m$ in Eq. 5 is the slope of a linear least square fit obtained between $a(R)$ and $R$, for each of the selected
frequencies. Further, the $Q$ values are estimated for the selected frequencies by substituting the value of $m$
computed using Eq. (7). Columns 2 and 3, Table 4 list the value of $m$ and $Q$ with frequency $(f)$ respectively. In
order to build the frequency dependent relationship $Q_s = Q_0 f^n$, the value of $Q$ is fitted as a function of
frequency using a power law. In the above expression, $n$ is the frequency dependent coefficient, which is





approximately equal to 1 and varies on the basis of the heterogeneity of the medium (Aki 1980). Variation of $Q$
against frequency as illustrated in Figure 4 gives frequency dependent $Q_s$ for the North-west Himalaya as;
$\quad Q_s = 105\, f^{0.94}$ (8)
The values of $n$ and $Q_0$ (in the expression $Q_s = Q_0\, f^n$) are attributed to the level of tectonic activity and degree
of heterogeneity respectively, present in the region. Aki (1980) concluded higher values of $n$ for tectonically
active regions in comparison to that of stable regions. Similarly, low value of $Q_0$ (<200) is an indication of
larger degree of heterogeneities in the medium (Joshi 2006). The values of $n$ (=0.94) and $Q_0$ (=104), obtained in
this study indicates that the present study region is tectonically active, characterized by higher degree of
heterogeneities, in accordance with Aki (1980) and Joshi (2006).
**3.4 Comparison with Regional and Global Attenuation Characteristics**
As discussed earlier, numerous studies exist where path attenuation of different parts of the present study area
were attempted in the past. Comparison of present results with those obtained by the previous researchers for the
NorthWest Himalaya and Delhi region are attempted as shown in Figure 5. It can be seen from Figure 5 that the
attenuation curve obtained in the present study falls in between existing attenuation curves for the North-west
Himalaya in the literature, [Kinnaur, (Kumar et al., 2009), Kumoan (Mukhopadhyay et al., 2010), Garhwal
regions (Negi et al., 2015) and Delhi (Sharma et al., 2015)]. It has to be highlighted here that the data base for
the present study includes EQ records from Kinnaur, Kumoan, Garhwal regions of North West Himalaya as well
as from regions around Delhi. For this reason, the value of $Q_0$ and $n$ obtained in the present study reflects an
average attenuation of regions encompassing North-west Himalaya up to Delhi region.
Furthermore, the attenuation results obtained in this study is compared with some typical results
obtained globally in terms of attenuation characteristics and tectonic setting as shown in Figure 6. Literature
suggests low values of $Q_s$ for tectonically active regions [e.g. Kato Japan region (Yoshimoto et al., 1993); East
central Iran (Mahood et al., 2009); Egypt (Abdel 2009); and Umbria–Marche region (Lorenzo et al., 2013)].
Similarly, relatively high values of $Q_s$ were found for tectonically stable areas [e.g. Baltic Shield (Kvamme and
Havskov 1989); Central south Korea (Kim et al., 2004) and South Eastern Korea (Chung and Sato 2001)]. The
attenuation values obtained in the present study show good agreement with other studies with lower value of $Q_s$.
Further, attenuation curves for the present region is found closer to regions of high seismicity like Umbria–
Marche and Eastern Iran as can be observed from Figure 6.
**4 Site Effects**
After estimation of path parameter as discussed in the previous section, site characteristics of recording stations
are determined using the second step of inversion. In addition to GINV, site components are also estimated
using HVSR method. Detailed discussion on GINV and HVSR is given in the subsequent sections.
**4.1 GINV**
The GINV was developed by Andrews (1986) by recasting the method of spectral ratio into a generalized
inversion problem. Since then, various forms of this technique have been developed and used for estimating the



seismic characteristics by various researchers (Castro et al., 1990; Boatwright et al., 1991; Oth et al., 2008 etc.).
The methodology used for estimating site characteristics in the present study is discussed here.
As per Iwata and Irikura (1988), the observed Fourier amplitude (acceleration) spectrum (FAS) of the
$i^{th}$ EQ recorded, at the $j^{th}$ recording station, $U(f)_{ij}$ can be represented in the frequency domain as the product
of source term $(S(f)_{ij})$, path attenuation $(A(f)_{ij})$ and site term $(G(f)_j)$ as shown below;
$$U(f)_{ij} = S(f)_{ij}\, A(f)_{ij}\, G(f)_j \qquad (9)$$

Further, the path attenuation term can be removed from the spectral content of the record following Andrews
(1986) as;
$$U^A(f)_{ij} = \frac{U(f)_{ij}}{A(f)_{ij}} = S(f)_{ij}\, G(f)_j \qquad (10)$$

The value of $A(f)_{ij}$ is estimated here using Eq. (3) and by considering $Q_s$ as per Eq. (8), obtained in the earlier
section. Eq. (10) can be linearized, by taking natural logarithms on both sides as per Andrews (1986) giving;
$$ln\,U^A(f)_{ij} = ln\,S(f)_i + ln\,G(f)_j \qquad (11)$$

Considering: $ln\,S_i = s_i(f)$, $ln\,G(f)_j = g(f)$ and $ln\,U^A(f)_{ij} = d_{iji}$, Eq. (11) in the matrix form can be
written in accordance with Joshi et al., (2010) and following the notations of Menke (1989) as;

| ← | 1st event | → | | ← | nth event | → | | ← Site effect→ | | | | | |
|---|---|---|---|---|---|---|---|---|---|---|---|---|---|
| 1 | 2 | … | m | 1 | 2 | … | m | 1 | 2 | … | m | | |
| 1 | 0 | … | 0 …… | 0 | 0 | | 0 | 1 | 0 | … | 0 | $s_1(f_1)$ | $d_1(f_1)$ |
| 0 | 1 | | 0 …… | 0 | 0 | | 0 | 0 | 1 | … | 0 | : | : |
| : | | | : | : | : | | : | : | : | | : | : | : |
| : | | | : | : | : | | : | : | : | | : | : | : |
| 0 | 0 | 0 | 1 | 0 | 0 | … | 0 | 0 | 0 | … | 1 | $s_1(f_n)$ | $d_1(f_m)$ |
| | | | | | | | | | | | | $s_n(f_1)$ = | |
| | | | | | | | | | | | | : | |
| For nth earthquake | | | | | | | | | | | | $s_n(f_n)$ | $d_n(f_m)$ |
| 0 | 0 | … | 0 …… | 1 | | … | 0 | 1 | 0 | … | 0 | $g(f_1)$ | $d_n(f_m)$ |
| 0 | 0 | … | 0 …… | 0 | 1 | … | 0 | 0 | 1 | … | 0 | $g(f_2)$ | : |
| : | : | : | : | : | : | | : | : | : | | : | : | : |
| : | : | : | : | : | : | | : | : | : | | : | : | : |
| 0 | 0 | … | 0 … | 0 | 0 | … | 1 | 0 | 0 | … | 1 | $g(f_m)$ | $d_n(f_m)$ |

(12)

The matrix form in Eq. (12) represents a purely indeterminate system since there are $(n + 1) \times m$ unknowns
for '$m \times n$' data (here $m$ is the number of sample frequency and $n$ is the number of EQs recorded at a
particular recording station). Further, Eq. (12) is solved using minimum norm inversion procedure similar to the



work by Joshi et al., (2010), Harinarayan and Kumar (2017) to determine $g(f)_j$ at each of the selected
recording stations.

Based on the above discussed methodology, inversions are performed for east-west, north-south and

vertical components of EQ records separately to obtain the amplification curves in the frequency range of
0.25Hz to 15Hz for each of the three components. For further calculation, the horizontal component is obtained
as the geometric mean of east-west and north-south components.
**4.2 HVSR**
HVSR method is an extension of Nakamura (1989) technique, which has been widely used in the recent years to
assess the subsoil characteristics using recorded ambient noises. Nakamura (1989) technique is based on the
assumption that the soil amplification effects are retained only in the horizontal component whereas the source
and the path effects are maintained both in vertical as well as horizontal components of ground motion. Hence,
the ratio of horizontal and vertical components gives an estimate of site amplification. Lermo and Chavez-
Garccia (1993) extended Nakamura (1989) technique to S wave part of the accelerograms and studied the
theoretical basis of the technique by numerical modelling of SV waves. Later, HVSR method was applied to EQ
recordings worldwide (Luzi et al., 2011; Yaghmaei-Sabegh and Tsang 2011; Alessandro et al., 2012;
Harinarayan and Kumar 2017, 2018 etc.) to obtain the site characteristics.
Comparative studies between HVSR and other methods of evaluating site parameters reported by Field and
Jacob (1995), Parolai et al., (2004), Shoji and Kamiyama (2002) Harinarayan and Kumar (2017) etc. show that,
HVSR can provide good and reliable estimate of predominant  frequency in the site amplification function.
However, the above literatures also point out discrepancies in amplification levels obtained from HVSR with
other methods. In order to compare the site amplification functions obtained from HVSR and GINV methods,
HVSR for each station are computed considering the same S wave window as used in the GINV method. In the
present work, HVSR for each recording station is determined using the following steps;
1.  Calculate the FAS for the three components (north-south, east-west and vertical) of ground motion records.
2.  Obtain the geometric mean of the two horizontal response spectra components (H) using  Eq. (13) given

below;

$H = (H_{EW} \times H_{NS})^{0.5}$                                                    (13)

3.  Calculate the ratio of $H$ to $V$ $(H/V)$.
Where, $H_{EW}$ and $H_{NS}$ are the FAS of the horizontal east-west and north-south components respectively and V is
the FAS of the corresponding vertical component. Then, the HVSR at each recording station can be determined
as;
$(HVSR)_i = \dfrac{\sum_{i=1}^{Ni} \frac{H}{V}}{N_i}$                                         (14)
Here, $N_i$ is the number of events recorded at recording station "$i$" and $(HVSR)_i$ indicates the average HVSR
value for a particular station "$i$". The $f_{peak}$ is the value of frequency corresponding to a maximum value of
$HVSR_i$ (denoted by $A_{peak}$) at the recording station "$i$".
**4.3 Site Parameters**





Site amplification curves are developed using GINV for the horizontal (GINV H) and the vertical components
(GINV V). Figure 7 shows typical amplification curves obtained for GINV H (indicated by dashed lines) and
GINV V (indicated by firm lines) at 6 stations. In general, obtained amplification value for GINV H is greater
than GINV V for all frequencies. A typical observation (from Figure 7) for both GINV H and GINV V is that
the high level of amplification is observed at high frequencies. For several recording stations, clear and distinct
peak in the amplification curve can be observed (eg. JAMI, BAR, GHA and SND) from Figure 7. Moreover, the
main peaks of the GINV V component are usually at higher frequencies than for the GINV H component.
Further, in the case of few recording stations, the shift in frequency is relatively close to a factor $\sqrt{2}$ (eg. BAR
and GHA). Next, Site amplification factor (SAF) is estimated based on the GINV results (denoted by GINV
H/V) as the ratio of GINV H to GINV V. The value of frequency corresponding to the maximum value of SAF
(denoted as $A_{peak}$) is $f_{peak}$. GINV H/V curves are compared with those estimated using HVSR method for a total
of 101 recording stations. Figure 8 shows the comparison of the HVSR (indicated by dashed lines) and GINV
H/V (indicated by firm line) for 9 recording stations that provide a good sample of typically observed effects for
all the recording stations in the present study. A general observation made from Figure 8 is that both HVSR and
GINV H/V show similar SAF patterns for all recoding stations in the studied frequency range. Overall value of
$f_{peak}$ obtained exhibit 1:1 matching between the two methods. However, there is trend of difference in terms of
$A_{peak}$ values. $A_{peak}$ values obtained using HVSR are found higher compared to those obtained using GINV H/V
curves. This observation was also reported by many studies in other regions (Sharma et al., 2014; Field and
Jacob, 1995). The values of $f_{peak}$ obtained using GINV H/V and HVSR are tabulated in Column 5 and 7, Table
1. Similarly, the values of $A_{peak}$ obtained using GINV H/V and HVSR are tabulated in Column 6 and 8, Table 1.
The maximum value of $f_{peak}$ of 15Hz is observed for the recording station GGI with a value of $A_{peak}$ of 5.3. The
maximum value of $A_{peak}$ of 12.2, based on GINV H/V is observed for ADIB recording station at 6.3Hz. The
range of $A_{peak}$ based on HVSR varies between 1.7 and 19.4, while based on GINV H/V, the range of $A_{peak}$ varies
between 1.5 and 12.7. The range of $f_{peak}$ based on HVSR varies between 0.4Hz and 10Hz, while based on GINV
H/V, the range of $f_{peak}$ varies between 0.5Hz and 15Hz.
Further, based on the value of $f_{peak}$ obtained using GINV, the recording stations are classified as either
rock sites or soil sites. In general, criteria based on average shear wave velocity over 30m ($V_{s30}$) is used for site
classification. A site can also be classified based on $f_{peak}$ values. Such an approach was used by Harinarayan and
Kumar (2017) to classify the recording stations in the North-West Himalaya based on $f_{peak}$ obtained using HVSR
method, where stations having $f_{peak}$ less than 6.35Hz were classified as soil sites and stations having $f_{peak}$ greater
than 6.35Hz were classified as rock sites. The range of $f_{peak}$ values reported by Harinarayan and Kumar (2017)
for soil and rock sites were calculated based on the range of $V_{s30}$ based on NEHRP site classification scheme in
accordance with the Eq. (15) (Kramer, 1996), correlating $f_{peak}$ to soil depth (denoted by H and taken as 30m)
and shear wave velocity ($V_z$).
$$f_{peak} = {V_z}/{4H}$$  (15)
Based on this classification criteria, all the recording stations in the present are classified as either rock site or
soil site and is given in Column 9 of Table 1. Out of 101 recording stations 10 recording stations are classified
as rock sites and the rest 91 stations are classified as soil sites.



**4.4 Spatial distribution of GINV H/V characteristics**
To provide further insight into the GINV results, distribution of $f_{peak}$ (Fig. 9) and $A_{peak}$ (Fig. 10) over the region
confined by the Delhi, Himachal Pradesh, Haryana, Uttarakhand and Punjab are developed separately.
Considering the distribution of $f_{peak}$ for the Delhi region (Figure 9A), an increasing trend from west to east with
a range of 2 to 3Hz in the western region and 3 to 4.67Hz in the eastern region is observed. Spatial distribution
of $A_{peak}$ for the Delhi region (Figure 10A) reveals $A_{peak}$ in the range 2 to 3 in the western region. For the state of
Haryana, the values of $f_{peak}$ and $A_{peak}$, in general, are found to be in the range of 1 to 5.3Hz and 2 to 4 (see
Figures 9B and 10B respectively). For the state of Himachal Pradesh spatial distribution of $f_{peak}$ (Fig 9C) shows
an increasing trend from south to north with a range of 3 to 4.67Hz in the northern region and 1.5 to 3Hz in the
southern region. The value of $A_{peak}$ for the Himachal Pradesh in the range 1.8 to 6.2. For the state of Punjab, the
spatial distribution of $f_{peak}$ (Figure 9D) shows a decreasing trend from east (3Hz) to west (1.5Hz) whereas, an
increasing trend is observed form the east (3 to 5) to west (2 to 3) in the case of $A_{peak}$ values. The range of $f_{peak}$
for the state of Uttarakhand varies from 0.5 to 5Hz, and $A_{peak}$ varies from 1.5 to 6.5 (Figure 9E and 10E). The
spatial variation of $A_{peak}$ for the region of Uttarakhand shows an increasing trend from north to south. The
spatial distribution of $A_{peak}$ and $f_{peak}$ discussed above and shown in Fig 9 and Fig. 10 can be useful for regional
seismic hazard analysis.
**4.5 Relationship between GINV H/V results and $V_{s30}$**
The value of $V_{s30}$ are available for 8 recording stations in Terai region of Uttarakhand and 19 stations in Delhi
region from Pandey et al., (2016a) and Pandey et al., (2016b) respectively based on results of MASW test. The
values of $V_{s30}$ for a total of 27 recording stations in Terai region of Uttarakhand and Delhi regions coincides
with recording stations where $A_{peak}$ and $f_{peak}$ are determined in the present study. The values of $V_{s30}$ (obtained
from Pandey et al., 2016 a, b) and $f_{peak}$ and $A_{peak}$ (as per present work) for above 27 recording stations are listed
in Table 5. Based on the present findings, relationship between $f_{peak}$ and $V_{s30}$ for the 27 recording stations is
proposed as shown in Figure 11a with an $R^2$ value of 0.71 as:
$$\log V_{s30} = (0.48)\left(\log f_{peak}\right) + (2.33) \tag{16}$$

Similarly, the relationship between $A_{peak}$ and $V_{s30}$ for above 27 recording stations, as obtained in the present
study is shown in Figure 11b is as follows:
$$\log V_{s30} = -(0.74)\left(\log A_{peak}\right) + (2.93) \tag{17}$$

A lack of correlation (correlation coefficient =0.47) between $V_{s30}$ and $A_{peak}$ is observed as shown in Figure 11B,
which is also reported in the previous studies like Dutta et al., (2001), (2003) and Hassani et al., (2011). It can
be concluded from the proposed correlations (Eqs. 16 and 17) that the value of $f_{peak}$ increases with increase in
$V_{s30}$ whereas the value of $A_{peak}$ decreases with the increase in the value of $V_{s30}$. It has to be highlighted here that
both the equations [Eqs. 16 and 17] are applicable for sites having $f_{peak}$ in the range 1.8 to 6 Hz, and $A_{peak}$ in the
range 2 to 6.9.




**Conclusion**


The strong motion recordings available in PESMOS databank from 2004 to 2016 for the North-west Himalaya
and its surrounding areas are separately analysed to determine the path attenuation and site parameters using a
two-step inversion procedure. In the first step of inversion, non-parametric attenuation curves have been
developed. The attenuation with hypocentral distance shows a kink around 105km indicating the presence of
reflected and refracted arrival from the Moho discontinuity. At hypocentral distance less than 105km, the
attenuation curves decreases monotonically with distance. The S wave quality factor for distances less than
105km is described well as a function of frequency as: $Q_s = 105 f^{0.94}$. The values of $n$ (=0.94) and $Q_0$ (=104)
obtained in the present study indicates the region to be heterogeneous and seismically active. The $Q_s$ obtained in
this study is comparable with those estimated in various regions of North West Himalaya and Delhi NCR
region. Further, the attenuation characteristics of S waves in the present study are found close to other similar
and seismically active regions of the world.
In the second step of inversion, amplification curves for horizontal and vertical components are
computed and SAF for all 101 recording stations are estimated. The value of $f_{peak}$ and $A_{peak}$ obtained from the
SAF curves are in the range of 0.5 to 15Hz and 1.5 to 12.7 respectively. SAF are also estimated using HVSR
method and general comparison between the two methods shows similarities in terms of the general shape and
the value of $f_{peak}$, even though there is difference in the values of $A_{peak}$. Further, the recording stations are
classified as rock site or soil site based on the values of $f_{peak}$. Out of 101 recording stations 10 stations are
classified as rock sites and 91 stations are classified as soil sites. Further, based on the values of $f_{peak}$ and $A_{peak}$
spatial distribution maps for the states of Delhi, Haryana, Himachal Pradesh, Uttarakhand and Punjab have been
developed separately.
In conclusion, the path and site parameters found in this study provide important elements for further
strong motion simulations and seismic hazard assessment. Identifying recording station on rock sites enables to
utilize PESMOS data base for inversion studies requiring reference site, especially for computing EQ source
parameters.

**Authors Contribution:**
Harinarayan N H developed code generalized inversion, analyzed the records and all relevant literature review.
Kumar Abhishek (AK) highlighted the importance of site characterization for PESMOS recording stations and
need for the study.
**Acknowledgement**
The authors would like to thank the INSPIRE Faculty program by the Department of Science and Technology
(DST), Government of India for the funding project ''Propagation path characterization and determination of in-
situ slips along different active faults in the Shillong Plateau'' ref. no. DST/INSPIRE/04/2014/002617 [IFA14-
ENG-104] for providing necessary funding and motivation for the present study.

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



**FIGURES**

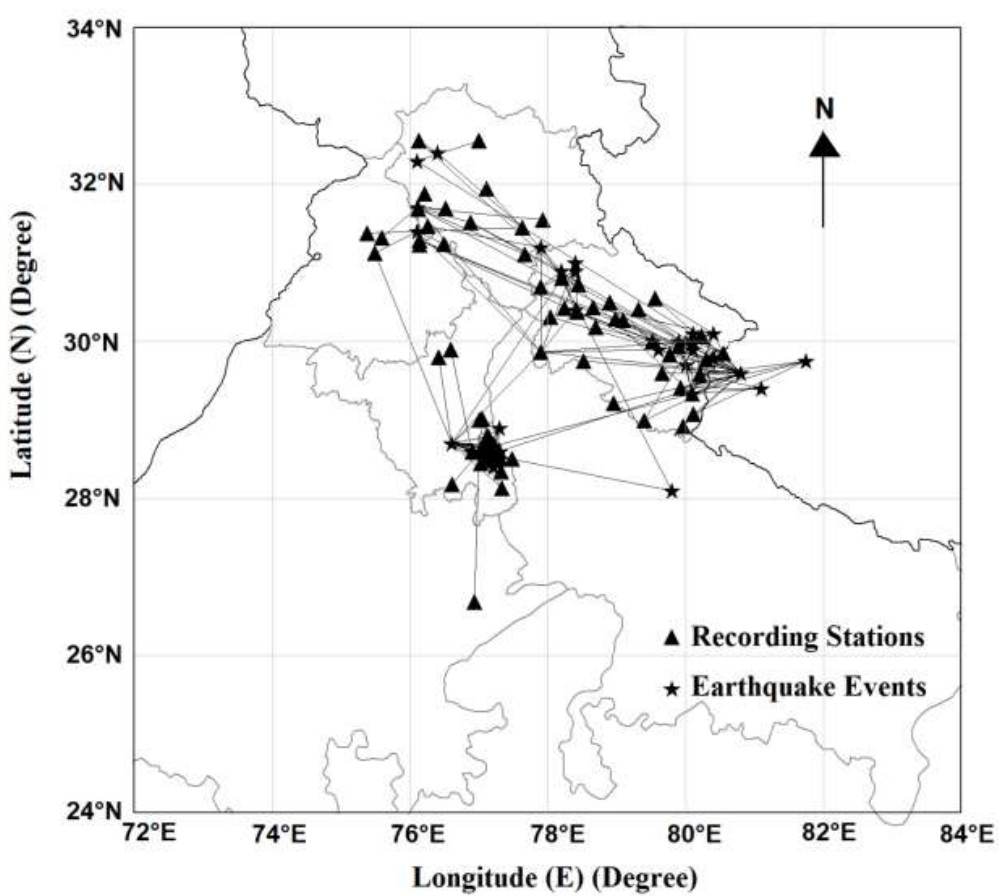

**Figure 1: Map of the region under study with EQs (stars), recording stations (triangles), and paths (solid-lines).**



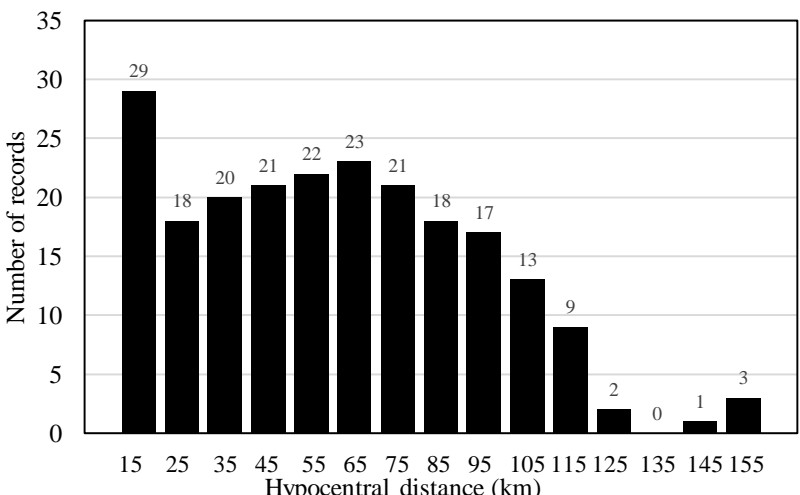


**Figure 2: Distribution of hypocentral distances in the data set.**

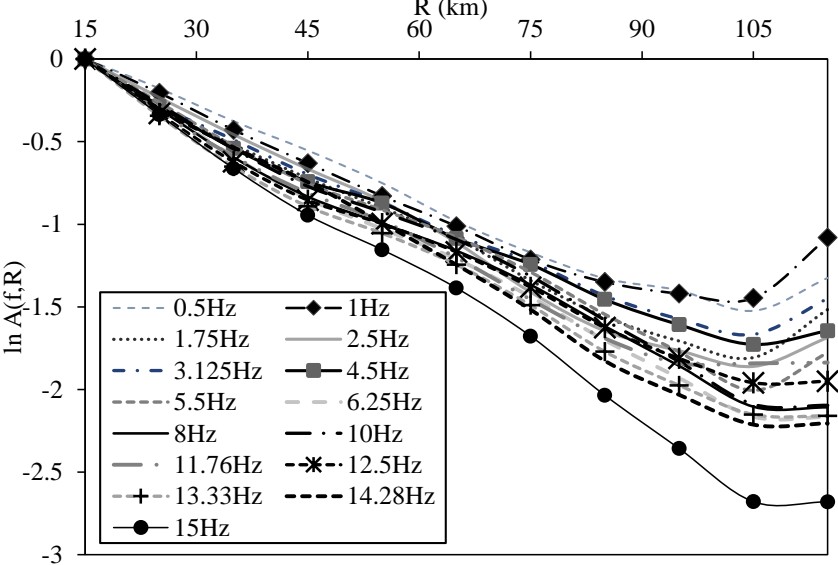


**Figure 3: S wave spectral attenuation versus hypocentral distance. Note that ln A(f,R₀) at reference distance is zero.**




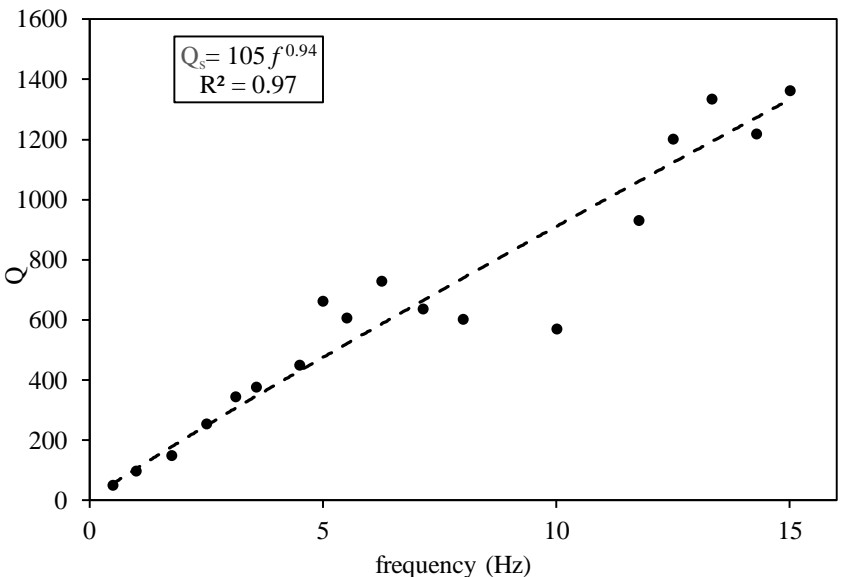


**Figure 4: Frequency dependence of the quality factor Q for hypocentral distances between 15 km to 105 km**

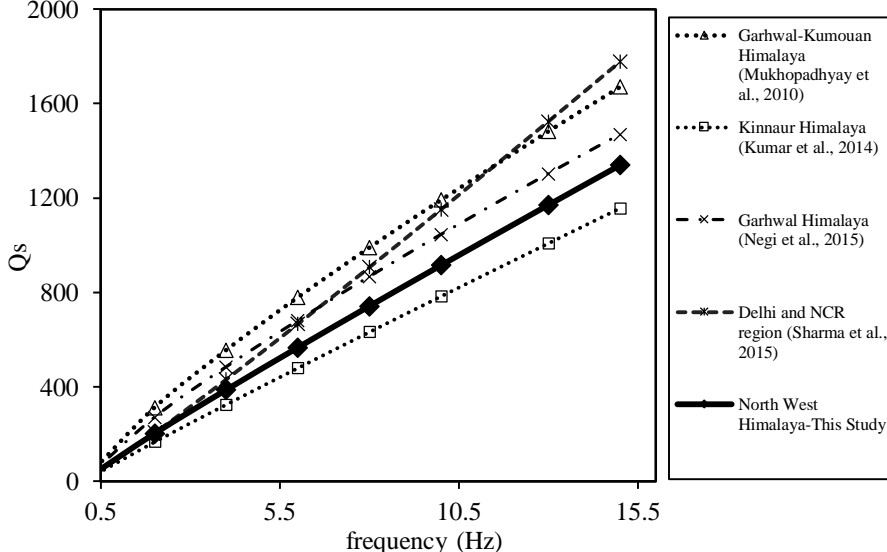


**Figure 5:  Comparison of Q$_S$ values of North West Himalaya with those obtained from parts of North West Himalaya**
**and Delhi region. The compared relations for Qs versus frequency are as follows: Garhwal-Kumouan Himalaya:**
$Q_S = 175 * f^{0.833}$ **(Mukhopadhyay et al., 2010); Kinnaur Himalaya:** $Q_S = 86 * f^{0.96}$ **(Kumar et al., 2014) ; Garhwal**
**Himalaya:** $Q_S = 151 * f^{0.84}$ **(Negi et al., 2015); Delhi and NCR region:** $Q_S = 98 * f^{1.07}$ **(Sharma et al., 2015).**





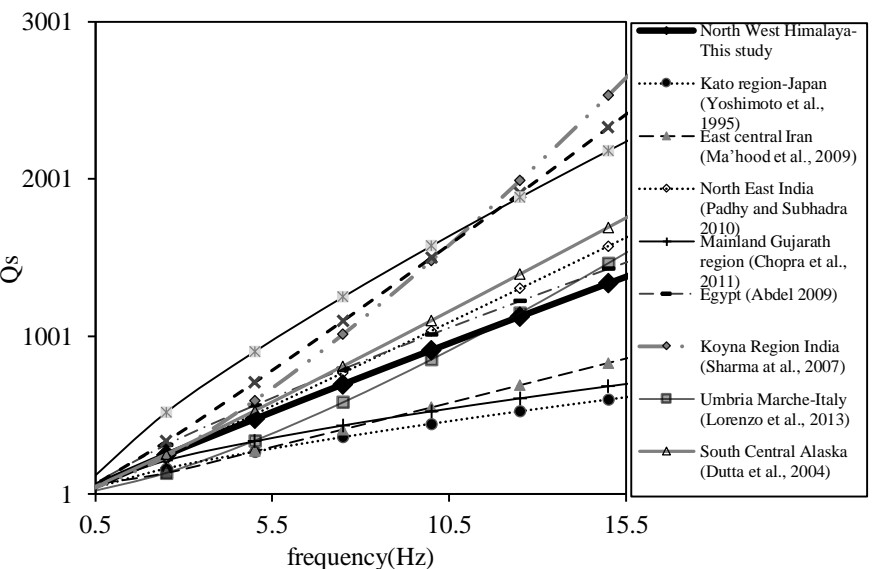


**Figure 6: Comparison of Q_S values of this study with regions of different tectonic settings of the world.**

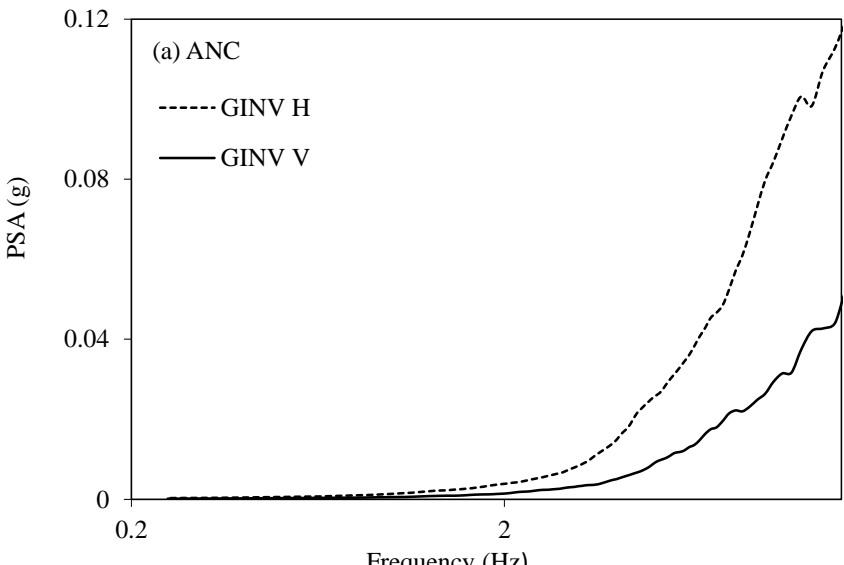






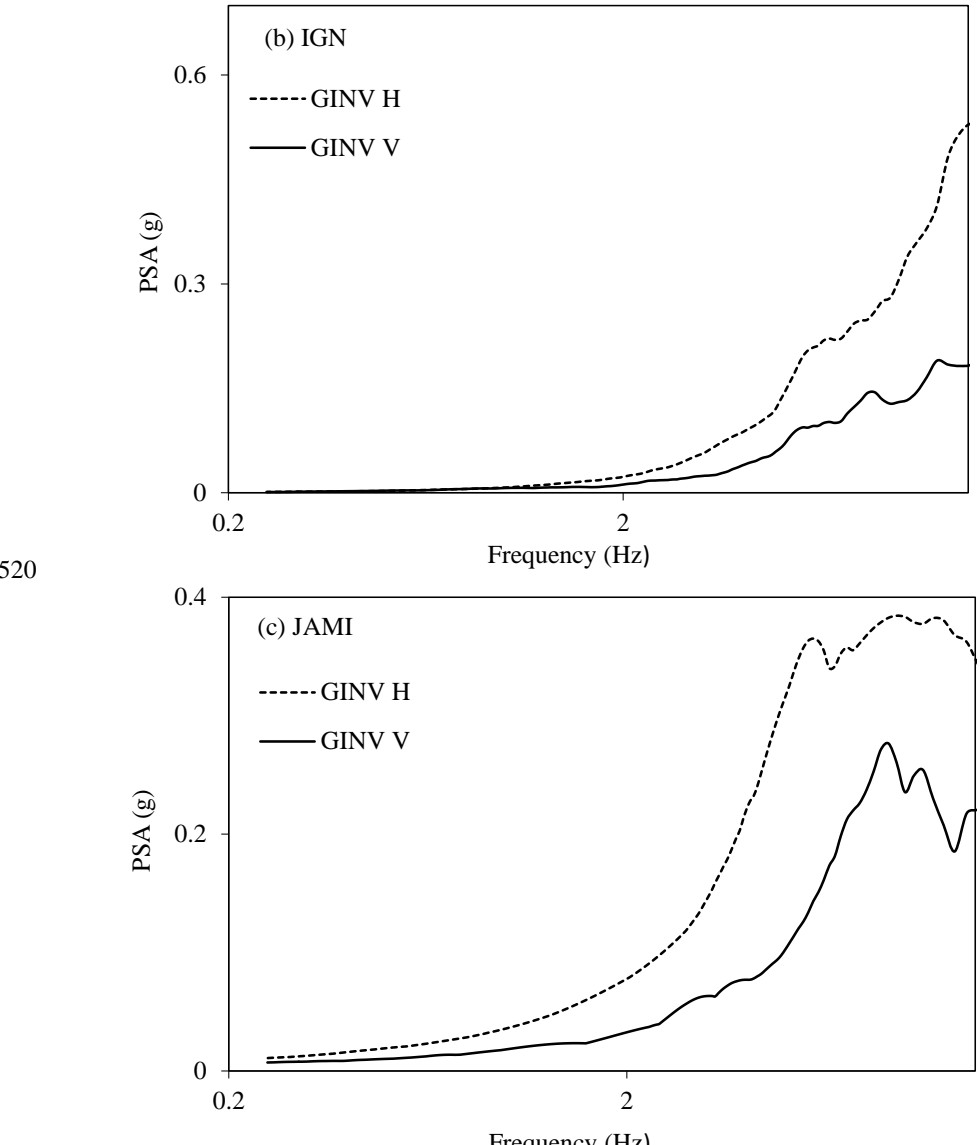







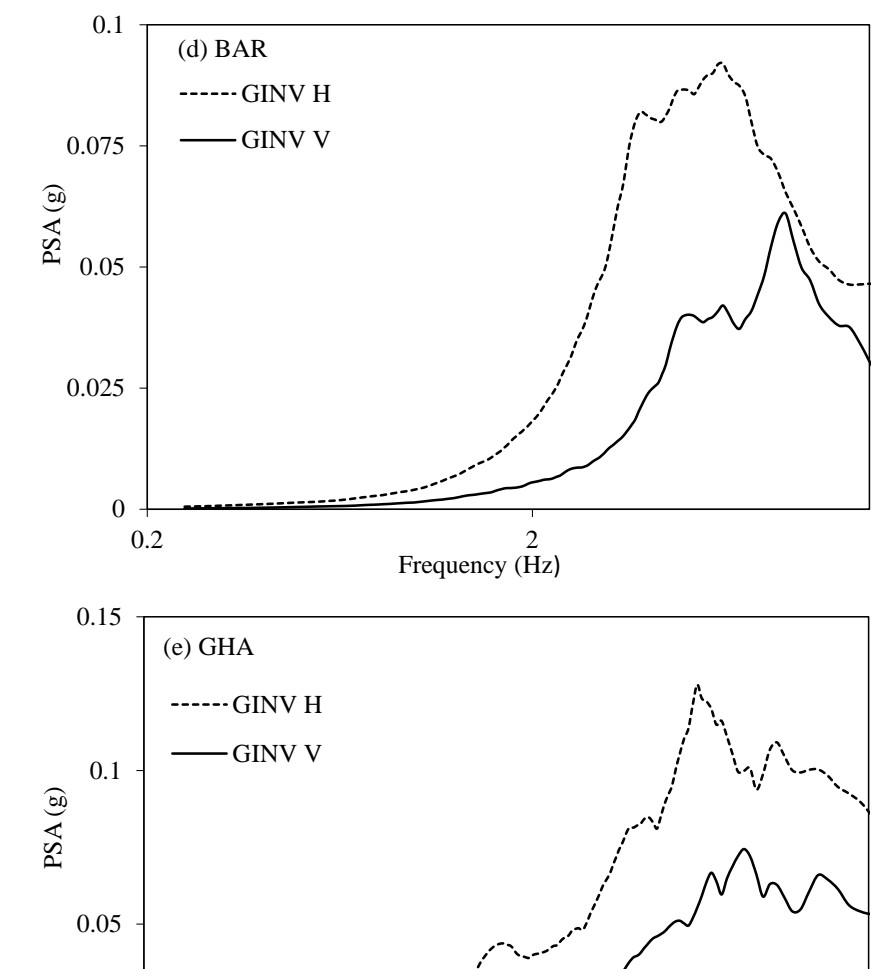







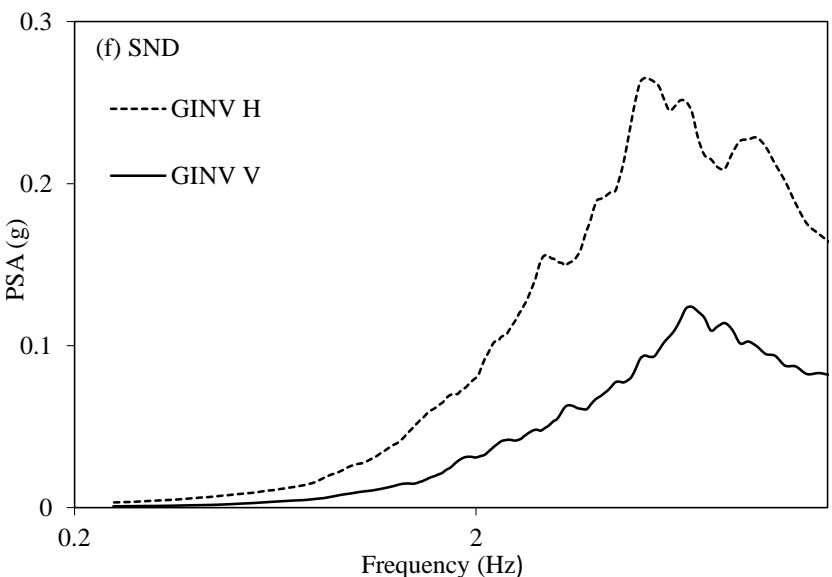


**Figure 7 (a-f). Site amplification curves obtained using GINV for horizontal component and vertical component**

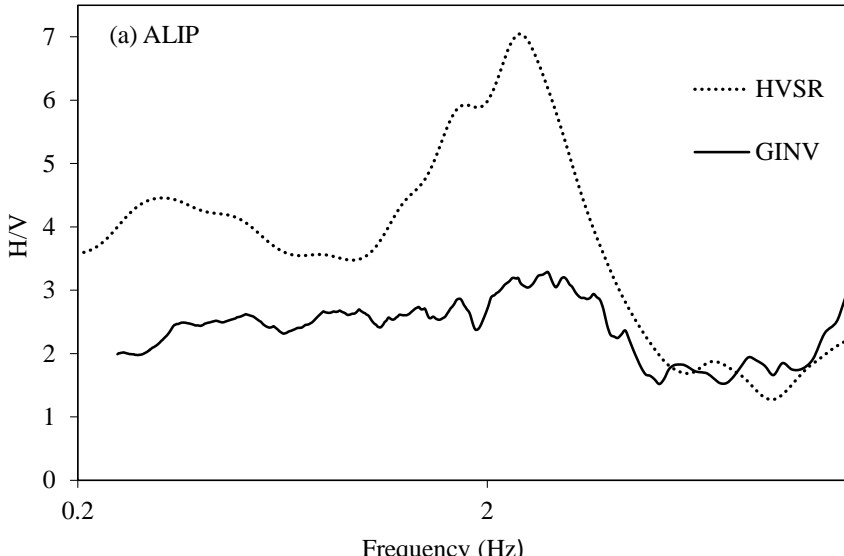




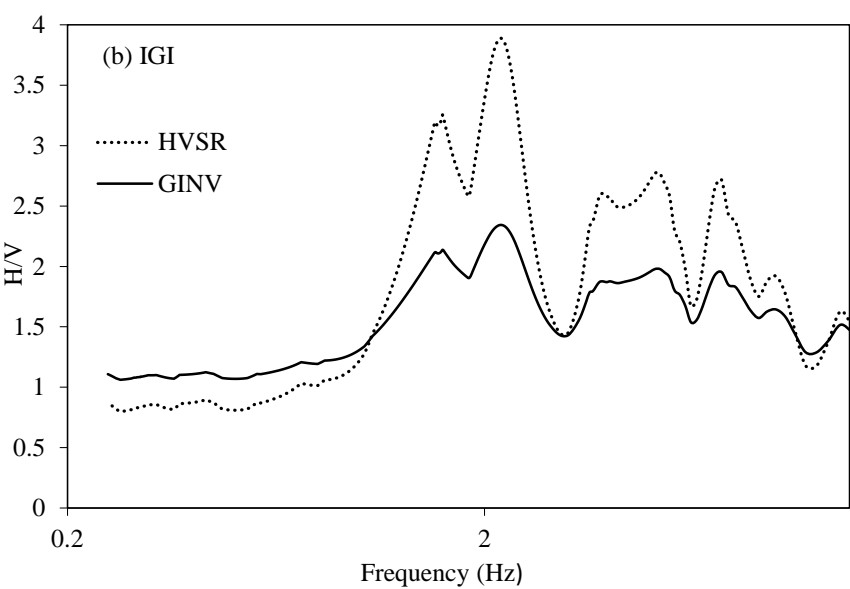


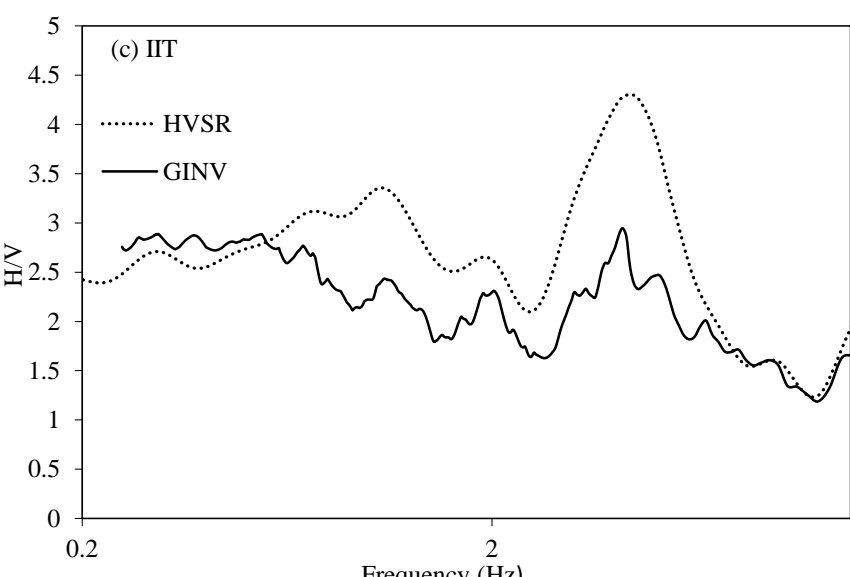




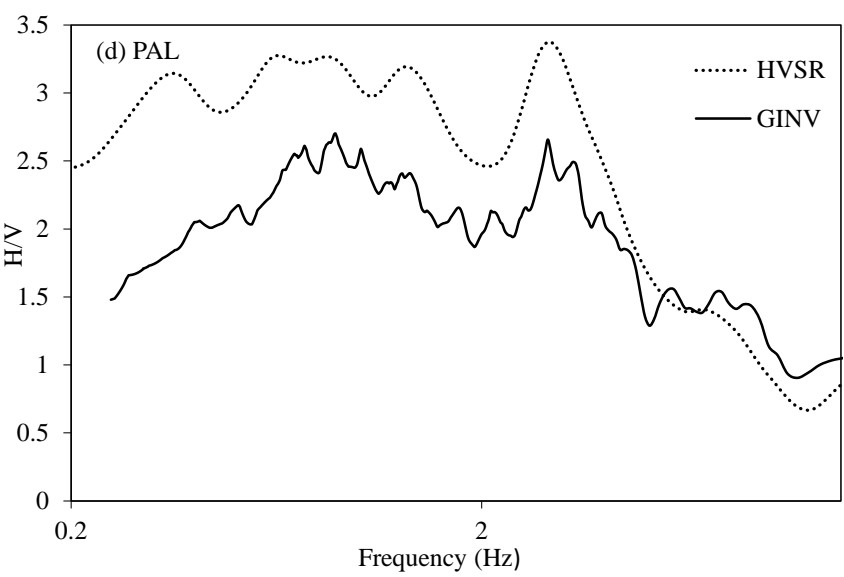


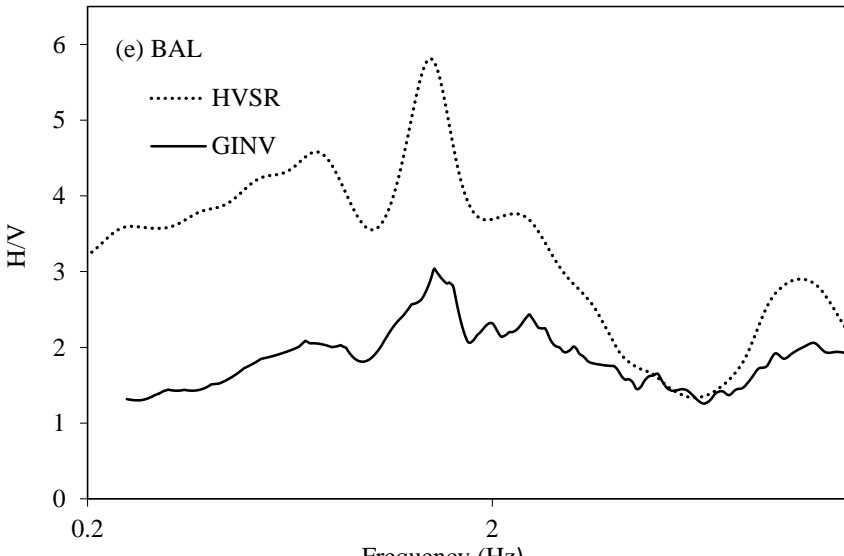





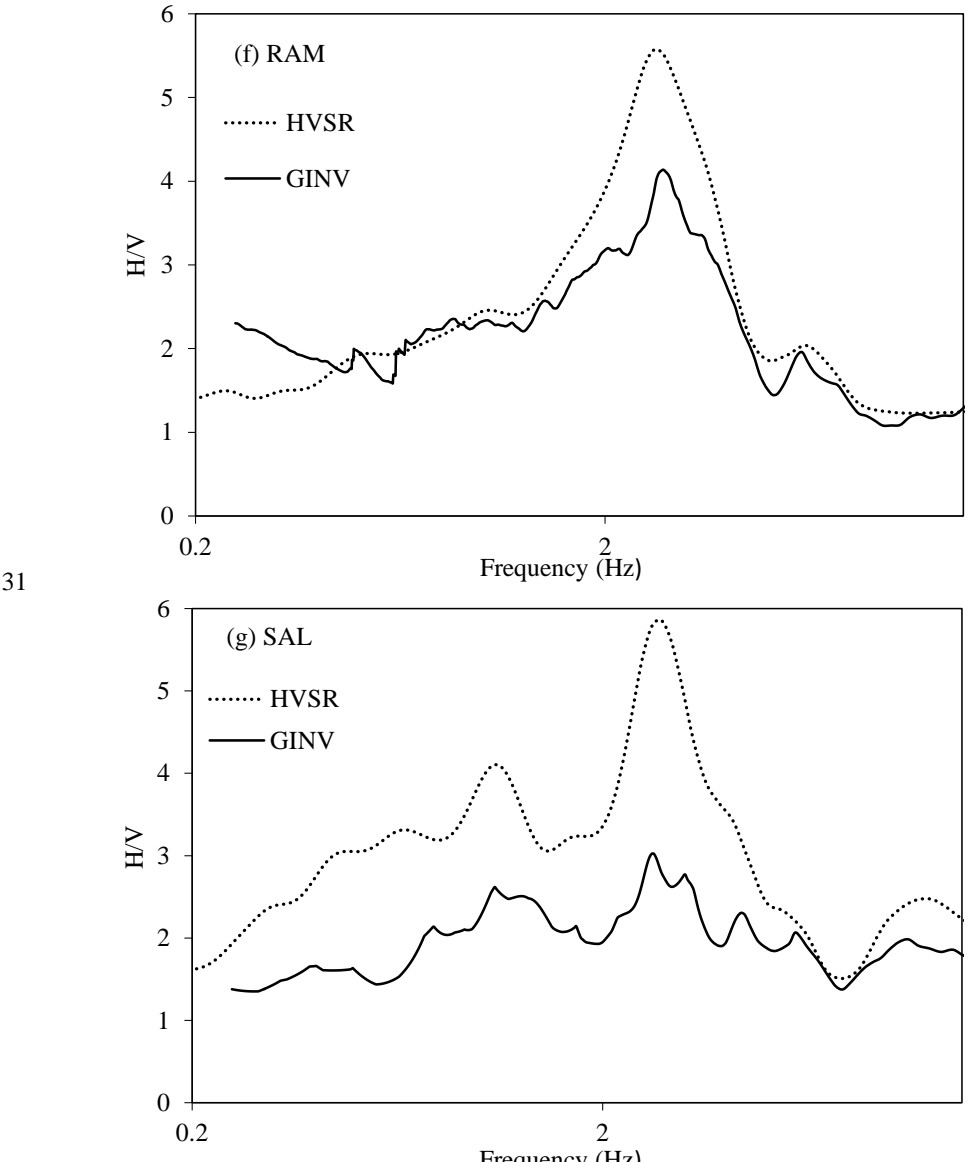







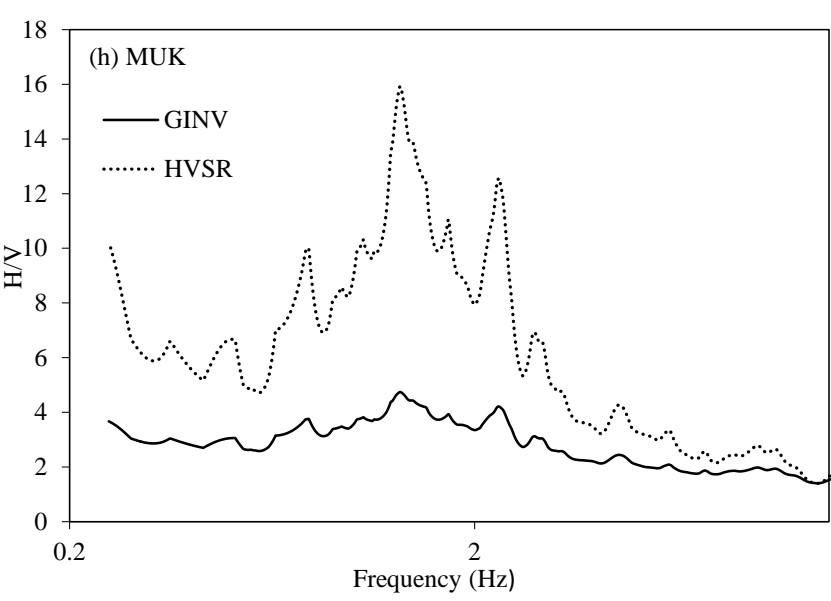



**Figure 8 (a-i): Horizontal to vertical ratio curve obtained using GINV and HVSR method.**











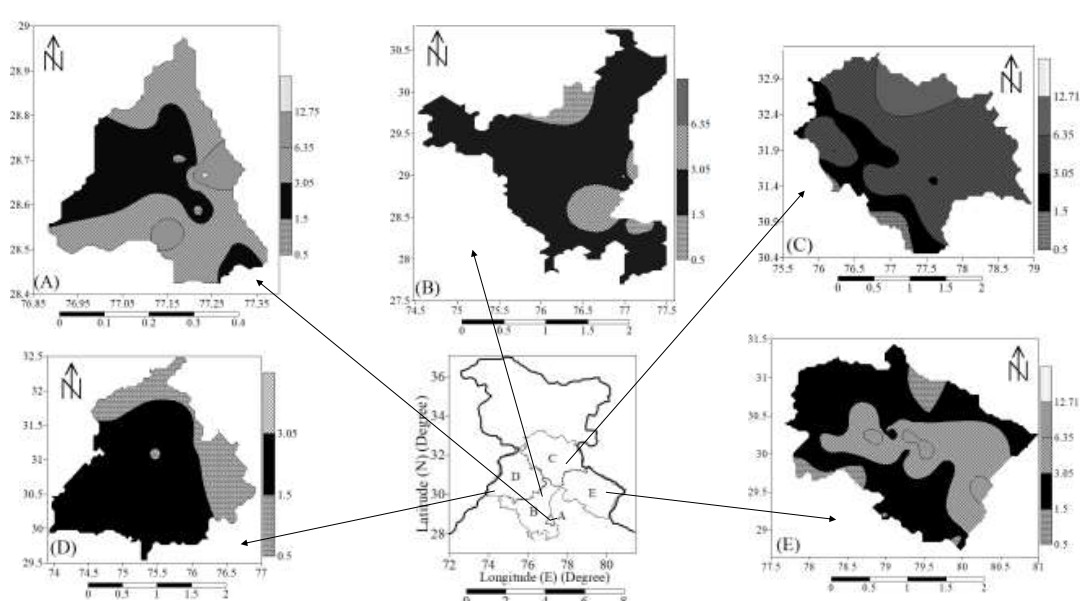


**Figure 9. Spatial distribution of estimated f$_{peak}$ for: (A) Delhi, (B) Haryana, (C) Himachal Pradesh, (D) Punjab and**
**(E) Uttarakhand**

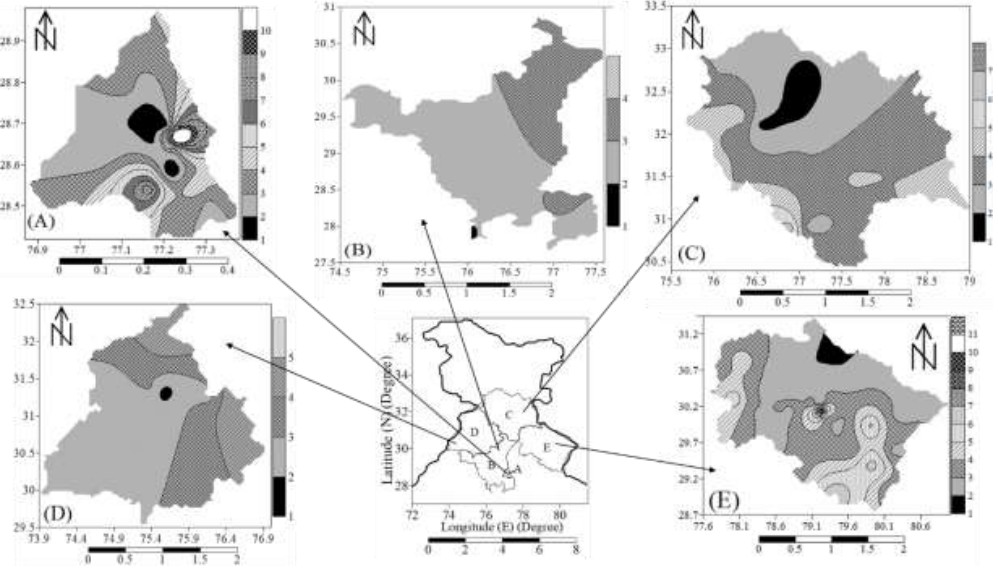


**Figure 10: Spatial distribution of estimated A$_{peak}$ for: (A) Delhi, (B) Haryana, (C) Himachal Pradesh, (D) Punjab and**
**(E) Uttarakhand**





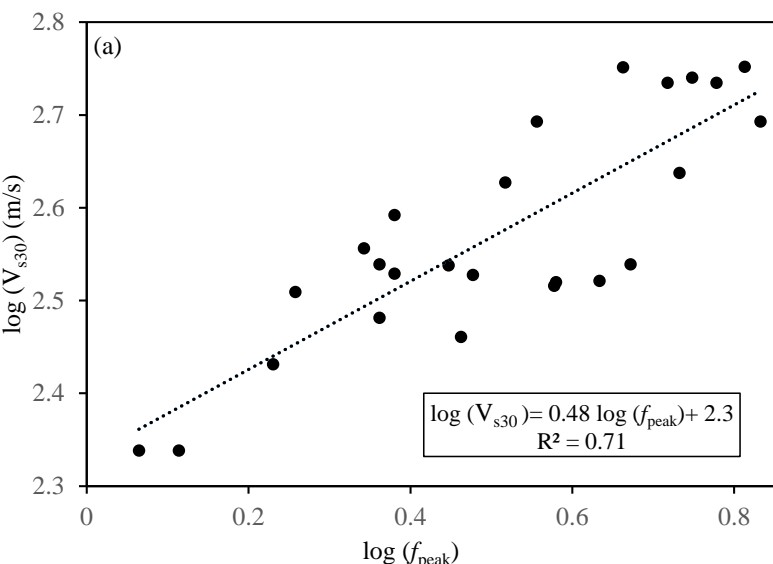


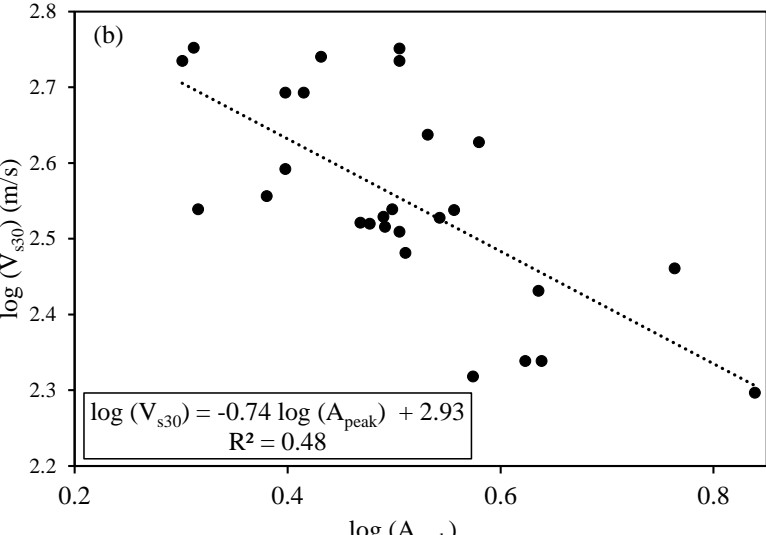


**Figure 11: $V_{s30}$ as a function of $f_{peak}$ (a) and $A_{peak}$ (b) of GINV method for recording stations at Delhi and Tarai region of Uttarakhand.**











**TABLES**
**Table 1: Detail of strong motion recording stations.**

| | | | | GINV | | HVSR | | R* or S# | | | | | | GINV | | HVSR | | R* or S# |
|---|---|---|---|---|---|---|---|---|---|---|---|---|---|---|---|---|---|---|
| Si. no | Station Code | Lat.(°) (N) | Lon. (°) (E) | $f_{peak}$ | $A_{peak}$ | $f_{peak}$ | $A_{peak}$ | | Si. no | Station Code | Lat.(°) (N) | Lon. (°) (E) | $f_{peak}$ | $A_{peak}$ | $f_{peak}$ | $A_{peak}$ | |
| -1 | -2 | -3 | -4 | -5 | -6 | -7 | -8 | -9 | -1 | -2 | -3 | -4 | -5 | -6 | -7 | -8 | -9 |
| | Himachal Pradesh | | | | | | | | | Uttarakhand | | | | | | | | |
| 1 | AMB | 31.7 | 76.1 | 1.7 | 4.3 | 1.2 | 9.3 | S | 1 | ALM | 29.6 | 79.7 | 2.1 | 3.0 | 2.8 | 4.4 | S |
| 2 | BHA | 31.6 | 77.9 | 4.5 | 4.0 | 4.1 | 4.4 | S | 2 | BAG | 29.8 | 79.8 | 1.5 | 4.7 | 1.5 | 5.2 | S |
| 3 | CHM | 30.4 | 79.3 | 1.4 | 5.4 | 1.5 | 7.5 | S | 3 | BAR | 30.8 | 78.2 | 3.0 | 4.5 | 2.8 | 7.0 | S |
| 4 | DEH | 31.9 | 76.2 | 6.8 | 3.5 | 10 | 5.4 | R | 4 | CHM | 32.6 | 76.1 | 3.6 | 2.4 | 2.0 | 2.9 | S |
| 5 | DHH | 32.2 | 76.3 | 2.7 | 4.5 | 2.7 | 4.9 | S | 5 | CHP | 29.3 | 80.1 | 5.4 | 5.2 | 5.6 | 6.5 | S |
| 6 | HAM | 31.7 | 76.5 | 2.9 | 3.3 | 3.1 | 6.6 | S | 6 | CKR | 30.7 | 77.9 | 2.1 | 3.8 | 2.0 | 4.5 | S |
| 7 | JUB | 31.1 | 77.7 | 5.8 | 3.3 | 5.6 | 4.9 | S | 7 | CMBB | 30.0 | 79.5 | 8.3 | 2.7 | 8.3 | 6.3 | R |
| 8 | KLG | 32.6 | 77.0 | 8.0 | 1.8 | 8.3 | 2.2 | R | 8 | DHA | 29.8 | 80.5 | 3.1 | 3.3 | 2.7 | 5.5 | S |
| 9 | KUL | 32.0 | 77.1 | 3.3 | 2.3 | 3.1 | 3.0 | S | 9 | DNL | 30.4 | 78.2 | 2.8 | 3.3 | 2.0 | 7.1 | S |
| 10 | MAN | 31.7 | 76.9 | 2.5 | 3.4 | 2.3 | 5.3 | S | 10 | DUN | 30.3 | 78.0 | 2.9 | 5.8 | 3.1 | 7.1 | S |
| 11 | RAM | 31.4 | 77.6 | 2.8 | 4.2 | 2.7 | 5.6 | S | 11 | GAR | 30.1 | 79.3 | 2.4 | 3.4 | 2.3 | 4.5 | S |
| 12 | SAL | 32.7 | 76.1 | 2.7 | 3.0 | 2.7 | 5.8 | S | 12 | GHA | 30.4 | 78.7 | 5.2 | 2.3 | 4.5 | 5.5 | S |
| 13 | SND | 31.5 | 76.9 | 5.0 | 3.5 | 5.0 | 4.2 | S | 13 | GLTR | 30.3 | 79.1 | 2.9 | 3.5 | 2.8 | 6.1 | S |
| 14 | SOL | 30.9 | 77.1 | 0.5 | 2.6 | 0.6 | 4.6 | S | 14 | JSH | 30.5 | 79.6 | 1.4 | 2.2 | 1.5 | 3.0 | S |
| 15 | UNA | 31.5 | 76.3 | 1.5 | 3.9 | 1.8 | 6.0 | S | 15 | KAP | 29.9 | 79.9 | 3.7 | 6.4 | 3.3 | 9.2 | S |
| 16 | KJK | 30.9 | 76.9 | 0.7 | 6.4 | 0.4 | 12.0 | S | 16 | KHA | 28.9 | 80.0 | 1.3 | 4.2 | 2.0 | 8.0 | S |
| 17 | PLM | 32.1 | 76.5 | 1.7 | 1.8 | 1.6 | 2.2 | S | 17 | KKHR | 30.2 | 78.9 | 9.5 | 3.5 | 9.5 | 9.3 | R |
| | Punjab | | | | | | | | 18 | KOT | 29.7 | 78.5 | 0.7 | 2.4 | 0.7 | 3.3 | S |
| 1 | ANS | 31.2 | 76.5 | 0.9 | 4.9 | 0.9 | 17.0 | S | 19 | KSK | 29.2 | 79.0 | 3.1 | 3.8 | 3.2 | 9.9 | S |
| 2 | ASR | 31.6 | 74.9 | 1.0 | 2.9 | 1.0 | 8.0 | S | 20 | KSL | 30.9 | 77.0 | 3.0 | 3.9 | 2.1 | 19.4 | S |
| 3 | GSK | 31.2 | 76.1 | 0.8 | 2.3 | 0.8 | 3.8 | S | 21 | LANG | 30.3 | 79.3 | 7.7 | 2.8 | 7.9 | 5.7 | R |
| 4 | JAL | 31.3 | 75.6 | 2.9 | 1.5 | 2.9 | 1.7 | S | 22 | LAN | 29.8 | 78.7 | 1.4 | 2.8 | 1.4 | 6.1 | S |
| 5 | KAT | 31.4 | 75.4 | 2.0 | 3.1 | 3.0 | 3.4 | S | 23 | MUN | 30.1 | 80.2 | 4.3 | 2.8 | 7.0 | 3.6 | S |
| 6 | MOG | 30.8 | 75.2 | 2.2 | 2.3 | 2.2 | 3.9 | S | 24 | PAU | 30.2 | 78.8 | 5.9 | 2.1 | 3.1 | 3.7 | S |
| 7 | MUK | 31.9 | 75.6 | 1.4 | 4.7 | 1.4 | 15.9 | S | 25 | PTH | 29.6 | 80.2 | 8.0 | 2.7 | 4.6 | 3.1 | R |
| 8 | NAW | 31.1 | 76.1 | 1.4 | 3.6 | 1.4 | 3.5 | S | 26 | PTI | 29.4 | 79.9 | 4.0 | 6.6 | 3.6 | 8.0 | S |
| 9 | NKD | 31.1 | 75.5 | 1.2 | 2.2 | 1.3 | 3.4 | S | 27 | RIS | 30.1 | 78.3 | 3.8 | 3.0 | 3.4 | 6.4 | S |
| 10 | PHG | 31.2 | 75.8 | 2.7 | 2.7 | 2.7 | 5.2 | S | 28 | ROO | 29.9 | 77.9 | 1.2 | 4.4 | 1.3 | 5.2 | S |
| 11 | TAR | 31.4 | 74.9 | 2.6 | 2.7 | 2.7 | 4.6 | S | 29 | RUD | 30.3 | 79.0 | 1.3 | 2.8 | 1.5 | 4.2 | S |
| | Delhi | | | | | | | | 30 | SMLI | 30.2 | 79.3 | 9.1 | 3.3 | 8.7 | 6.1 | R |
| 1 | ARI | 26.1 | 77.5 | 2.7 | 3.2 | 2.5 | 7.0 | S | 31 | TAN | 29.1 | 80.1 | 5.4 | 3.4 | 5.0 | 6.3 | S |
| 2 | IGN | 28.5 | 77.2 | 3.6 | 2.6 | 4.5 | 3.9 | R | 32 | THE | 30.4 | 78.4 | 1.6 | 2.8 | 1.5 | 3.6 | S |
| 3 | JNU | 28.5 | 77.2 | 9.0 | 2.1 | 8.7 | 3.5 | R | 33 | UDH | 29.0 | 79.4 | 2.7 | 6.9 | 2.2 | 10.1 | S |
| 4 | DJB | 28.7 | 77.2 | 2.2 | 3.2 | 10 | 4.5 | S | 34 | UTK | 30.7 | 78.4 | 2.4 | 2.9 | 2.3 | 4.4 | S |
| 5 | NDI | 28.7 | 77.2 | 6.8 | 2.5 | 7.2 | 3.7 | R | 35 | VIK | 30.5 | 77.8 | 2.3 | 3.8 | 2.3 | 10.4 | S |
| 6 | IMD | 28.7 | 77.2 | 6.0 | 2.0 | 6.3 | 2.9 | S | 36 | GDRI | 30.2 | 78.7 | 6.0 | 3.4 | 5.1 | 4.8 | S |
| 7 | NTPC | 28.5 | 77.3 | 2.8 | 3.6 | 2.8 | 5.4 | S | 37 | TLWR | 30.3 | 79.0 | 1.1 | 2.1 | 1.0 | 4.6 | S |
| 8 | ANC | 28.5 | 77.3 | 4.6 | 3.2 | 4.5 | 4.5 | S | 38 | UKMB | 30.3 | 79.1 | 1.0 | 2.9 | 1.4 | 10.0 | S |
| 9 | JAMI | 28.6 | 77.3 | 4.7 | 3.2 | 4.5 | 7.3 | S | 39 | ADIB | 30.2 | 79.2 | 6.4 | 12.7 | 6.3 | 17.8 | R |



| 10 | LDR | 28.6 | 77.2 | 0.7 | 4.3 | 0.9 | 7.0 | S | 40 | NUTY | 30.2 | 79.2 | 4.8 | 2.4 | 4.7 | 4.3 | S |
| 11 | VCD | 28.6 | 77.2 | 4.6 | 2.7 | 4.6 | 3.6 | S | 41 | KHIB | 30.2 | 78.8 | 7.7 | 3.3 | 8.0 | 8.7 | R |
| 12 | IIT | 28.6 | 77.3 | 4.3 | 2.9 | 4.5 | 4.3 | S | 42 | STRK | 30.3 | 79.0 | 4.7 | 2.8 | 4.7 | 5.8 | S |
| 13 | NSIT | 28.6 | 77.0 | 2.4 | 2.5 | 2.3 | 3.9 | S | 43 | NANP | 30.3 | 79.3 | 3.9 | 3.5 | 3.8 | 9.1 | S |
| 14 | RGD | 28.7 | 77.1 | 2.3 | 2.1 | 2.9 | 3.8 | S | | Haryana | | | | | | | |
| 15 | GGI | 28.7 | 77.2 | 15 | 5.3 | 15 | 8.4 | R | 1 | PAL | 28.1 | 77.3 | 2.8 | 2.7 | 2.9 | 3.4 | S |
| 16 | DLU | 28.7 | 77.2 | 1.8 | 3.2 | 1.9 | 3.7 | S | 2 | JAFR | 28.6 | 76.9 | 6.0 | 2.0 | 7.1 | 2.6 | S |
| 17 | DCE | 28.8 | 77.1 | 3.8 | 3.1 | 4.7 | 4.2 | S | 3 | GUR | 28.4 | 77.0 | 1.0 | 4.1 | 1.0 | 5.2 | S |
| 18 | IGI | 28.6 | 77.1 | 2.2 | 2.4 | 2.2 | 3.8 | S | 4 | REW | 28.2 | 76.6 | 2.5 | 2.1 | 2.5 | 3.7 | S |
| 19 | ZAKI | 28.6 | 77.2 | 3.9 | 3.5 | 3.9 | 8.4 | S | 5 | SON | 29.0 | 77.0 | 1.0 | 3.5 | 2.8 | 4.0 | S |
| 20 | ALIP | 28.8 | 77.1 | 2.3 | 3.2 | 2.5 | 6.9 | S | 6 | ROH | 28.6 | 77.2 | 1.4 | 3.1 | 2.0 | 4.6 | S |
| 21 | ROI | 28.6 | 77.2 | 1.4 | 3.1 | 2.0 | 4.6 | S | 7 | CRRI | 29.0 | 77.1 | 4.3 | 3.5 | 4.4 | 9.3 | S |
| | | | | R* | Rock site | | | | 8 | BAL | 28.3 | 77.3 | 1.5 | 3.0 | 1.4 | 5.8 | S |
| | | | | S# | Soil site | | | | 9 | KAI | 29.8 | 76.4 | 1.2 | 3.0 | 1.2 | 6.5 | S |


**Table 2: Details of earthquakes considered for estimation of site parameters in this work.**

| Event No. | dd/mm/yyyy | Lat. | Long. | Depth | Magnitude | Event No. | dd/mm/yyyy | Lat. | Long. | Depth | Magnitude |
|---|---|---|---|---|---|---|---|---|---|---|---|
| -1 | -6 | -2 | -3 | -4 | -5 | -1 | -6 | -2 | -3 | -4 | -5 |
| 1 | 14-12-2005 | 30.9 | 79.3 | 25.7 | 5.2 | 44 | 24-09-2011 | 30.9 | 78.3 | 10.0 | 3.0 |
| 2 | 07-05-2006 | 28.7 | 76.6 | 20.2 | 4.1 | 45 | 26-10-2011 | 31.5 | 76.8 | 5.0 | 3.5 |
| 3 | 29-11-2006 | 27.6 | 76.7 | 13.0 | 3.9 | 46 | 16-01-2012 | 29.7 | 78.9 | 10.0 | 3.6 |
| 4 | 10-12-2006 | 31.5 | 76.7 | 33.0 | 3.5 | 47 | 12-03-2012 | 28.9 | 77.3 | 5.0 | 3.5 |
| 5 | 22-07-2007 | 29.9 | 77.9 | 33.0 | 5.0 | 48 | 26-02-2012 | 29.6 | 80.8 | 10.0 | 4.3 |
| 6 | 25-11-2007 | 28.6 | 77.0 | 20.3 | 4.3 | 49 | 27-03-2012 | 26.1 | 87.8 | 12.0 | 3.5 |
| 7 | 04-10-2007 | 32.5 | 76.0 | 10.0 | 3.8 | 50 | 05-03-2012 | 28.7 | 76.6 | 14.0 | 4.9 |
| 8 | 18-10-2007 | 28.3 | 77.6 | 5.6 | 3.6 | 51 | 28-07-2012 | 29.7 | 80.7 | 10.0 | 4.5 |
| 9 | 19-08-2008 | 30.1 | 80.1 | 15.0 | 4.3 | 52 | 23-08-2012 | 28.4 | 82.7 | 5.0 | 5.0 |
| 10 | 19-10-2008 | 29.1 | 76.9 | 7.0 | 3.2 | 53 | 02-10-2012 | 32.4 | 76.4 | 10.0 | 4.9 |
| 11 | 21-10-2008 | 31.5 | 77.3 | 10.0 | 4.5 | 54 | 03-10-2012 | 32.4 | 76.3 | 10.0 | 3.6 |
| 12 | 31-01-2009 | 32.5 | 75.9 | 10.0 | 3.7 | 55 | 06-11-2012 | 32.3 | 76.2 | 5.0 | 4.1 |
| 13 | 09-01-2009 | 31.7 | 78.3 | 16.0 | 3.8 | 56 | 11-11-2012 | 29.3 | 80.1 | 5.0 | 5.0 |
| 14 | 25-02-2009 | 30.6 | 79.3 | 10.0 | 3.7 | 57 | 15-11-2012 | 30.2 | 80.1 | 5.0 | 3.0 |
| 15 | 18-03-2009 | 30.9 | 78.2 | 10.0 | 3.3 | 58 | 27-11-2012 | 30.2 | 78.4 | 12.0 | 4.8 |
| 16 | 04-09-2008 | 30.1 | 80.4 | 10.0 | 5.1 | 59 | 19-12-2012 | 28.6 | 76.8 | 10.0 | 2.9 |
| 17 | 01-05-2009 | 29.9 | 80.1 | 10.0 | 4.6 | 60 | 02-01-2013 | 29.4 | 81.1 | 10.0 | 4.8 |



| No | Date | | | | | No | Date | | | | |
|---|---|---|---|---|---|---|---|---|---|---|---|
| 18 | 15-05-2009 | 30.5 | 79.3 | 15.0 | 4.1 | 61 | 09-01-2013 | 29.8 | 81.7 | 5.0 | 5.0 |
| 19 | 17-07-2009 | 32.3 | 76.1 | 39.3 | 3.7 | 62 | 10-01-2013 | 30.1 | 80.4 | 5.0 | 3.2 |
| 20 | 27-08-2009 | 30.0 | 80.0 | 14.0 | 3.9 | 63 | 29-01-2013 | 30.0 | 81.6 | 7.0 | 4.0 |
| 21 | 21-09-2009 | 30.9 | 79.1 | 13.0 | 4.7 | 64 | 11-02-2013 | 31.0 | 78.4 | 5.0 | 4.3 |
| 22 | 03-10-2009 | 30.0 | 79.9 | 15.0 | 4.3 | 65 | 17-02-2013 | 30.9 | 78.4 | 10.0 | 3.2 |
| 23 | 06-12-2009 | 35.8 | 77.3 | 60.0 | 5.3 | 66 | 01-05-2013 | 33.1 | 75.8 | 15.0 | 5.8 |
| 24 | 11-01-2010 | 29.7 | 80.0 | 15.0 | 3.9 | 67 | 05-09-2013 | 30.9 | 78.5 | 11.0 | 3.5 |
| 25 | 22-02-2010 | 30.0 | 80.1 | 2.0 | 4.7 | 68 | 11-11-2013 | 28.5 | 77.2 | 10.0 | 3.1 |
| 26 | 24-02-2010 | 28.6 | 76.9 | 17.0 | 2.5 | 69 | 11-11-2013 | 28.4 | 77.2 | 11.0 | 2.8 |
| 27 | 14-03-2010 | 31.7 | 76.1 | 29.0 | 4.6 | 70 | 11-11-2013 | 28.4 | 77.2 | 12.0 | 2.5 |
| 28 | 03-05-2010 | 30.4 | 78.4 | 8.0 | 3.5 | 71 | 11-11-2013 | 28.4 | 77.2 | 13.0 | 3.1 |
| 29 | 28-05-2010 | 31.2 | 77.9 | 43.0 | 4.8 | 72 | 16-04-2013 | 28.0 | 62.1 | 16.0 | 7.8 |
| 30 | 31-05-2010 | 30.0 | 79.8 | 10.0 | 3.6 | 73 | 04-06-2013 | 32.7 | 76.7 | 18.0 | 4.8 |
| 31 | 06-07-2010 | 29.8 | 80.4 | 10.0 | 5.1 | 74 | 05-06-2013 | 32.8 | 76.3 | 10.0 | 4.5 |
| 32 | 10-07-2010 | 29.9 | 79.6 | 10.0 | 4.1 | 75 | 09-07-2013 | 32.9 | 78.4 | 10.0 | 5.1 |
| 33 | 26-01-2011 | 29.0 | 77.2 | 10.0 | 3.2 | 76 | 13-07-2013 | 32.2 | 76.3 | 10.0 | 10.0 |
| 34 | 14-03-2011 | 30.5 | 79.1 | 8.0 | 3.3 | 77 | 15-07-2013 | 32.6 | 76.7 | 30.0 | 4.4 |
| 35 | 18-02-2011 | 28.6 | 77.3 | 5.0 | 2.3 | 78 | 02-08-2013 | 33.5 | 75.5 | 20.0 | 5.4 |
| 36 | 09-02-2011 | 30.9 | 78.2 | 10.0 | 5.0 | 79 | 29-08-2013 | 31.4 | 76.1 | 10.0 | 4.7 |
| 37 | 04-04-2011 | 29.6 | 80.8 | 10.0 | 5.7 | 80 | 20-10-2013 | 35.8 | 77.5 | 80.0 | 5.5 |
| 38 | 15-06-2011 | 30.6 | 80.1 | 10.0 | 3.6 | 81 | 25-12-2013 | 31.2 | 78.3 | 10.0 | 4.0 |
| 39 | 20-06-2011 | 30.5 | 79.4 | 12.0 | 4.6 | 82 | 17-06-2014 | 32.2 | 76.1 | 10.0 | 4.1 |
| 40 | 23-06-2011 | 30.0 | 80.5 | 5.0 | 3.2 | 83 | 21-08-2014 | 32.3 | 76.5 | 10.0 | 5.0 |
| 41 | 28-07-2011 | 33.3 | 76.0 | 21.0 | 4.4 | 84 | 29-11-2015 | 30.6 | 79.6 | 15.0 | 4.0 |
| 42 | 07-09-2011 | 28.6 | 77.0 | 8.0 | 4.2 | 85 | 25-09-2016 | 30.0 | 79.5 | 11.0 | 3.7 |
| 43 | 21-09-2011 | 30.9 | 78.3 | 10.0 | 3.1 | 86 | 01-12-2016 | 30.6 | 79.6 | 19.0 | 4.0 |


**Table 3: List of Earthquakes and the corresponding stations considered for the estimation of path parameter.**

| Earthquake Event | Stations |
|---|---|
| 25-11- 2007 | HGR, NDI, CRRI, PAL, REW, NDI , CRRI, LDR, JAFR, IIT |
| 19-08-2008 | CHP, PTH, KAP, MUN |





| 04-09- 2008 | MUN, CHP, PTH, DHA , KAP, GHA, JSH |
|---|---|
| 01-05-2009 | MUN, BAG, KAP, GAR, CHM |
| 17-07-2009 | DHA, KLG |
| 27-08-2009 | KAP, BAG, MUN |
| 03-08-2009 | KAP, CHP, BAG |
| 11-01-2010 | PTH, CHP, DHA |
| 22-02-2010 | KAP, BAG, DHA, ROO, UDH |
| 24-02-2010 | RGD, IGN, ROH, DJB, CHP, ANC, JAMI, GGI, DLU, DCE |
| 14-03-2010 | DEH, JUB, SND, BHA, HAM, GAR, JAL, KAP, AMB, ROO |
| 03-04-2010 | THE, BAR, DNL, ROO |
| 28-05-2010 | JUB, BAR, ROO, UNA |
| 06-06-2010 | MUN, CHP |
| 10-07-2010 | BAG, KAP, GAR, ROO |
| 18-02-2011 | DJB, ANC |
| 09-02-2011 | UTK, SND, KUL, CKR |
| 04-04-2011 | JSH, CHP, PTI, PTH, ALM, DDH, BAG, DHA, GAR, MUN, RUD, THE, CHM, BAR, SND, KOT, DNL, LDR, ROO, TAN, KHA, UDH, KSH, DUD |
| 12-03-2012 | GGI, ANC, DLU, DCE |
| 27-03-2012 | ARI, ANC |
| 05-03-2012 | JAFR, JNU, DJB, IMD, PLW, GUR, NOI, NTPC, ANC, IIT, NSIT, ZAKI, ROO, RGD, GGI, DLU, DCE, ALIP, SON, BAR, KAI, NKD, |
| 02-10-2012 | CHA, RAM |
| 11-11-2012 | CHA, CHP, PTH |
| 27-11-2012 | UTK, THE, DNL, CKR |
| 02-01-2013 | CHP, PTI, PTH |
| 09-01-2013 | CHP, PTI, PTH, TAN |
| 11-02-2013 | UTK, ROO |
| 11-11-2013 (19:11:18) | NTPC, IGN, JNU, DJB, IMD, VCD, IGI, RGD, GGI, DLU, DCE, ALIP |
| 11-11-2013 (22,10,42) | IGN, JNU, DJB, VCD, RGD, DCE, ALIP |
| 11-11-2013 (20:11:30) | IGN, JNU, DJB, VCD, RGD, GGI, DLU, DCE |
| 29-08- 2013 | GSK, RAM, ROO, NKD, ANS, KAT |
| 25-09-2016 | UKMB, CMBB, GDRI, DURD |



**Table 4: Resulting parameters of eq. 7.**

| | | $Q_0 = (\pi f)/(\beta m)$ |
|---|---|---|
| f (Hz) | m | |
| (1) | (2) | (3) |
| 0.50 | 0.0095 | 51.65 |
| 1.00 | 0.0101 | 97.15 |
| 1.75 | 0.0115 | 149.32 |
| 2.50 | 0.0096 | 255.53 |
| 3.12 | 0.0089 | 344.54 |
| 3.57 | 0.0093 | 376.67 |
| 4.50 | 0.0098 | 450.57 |
| 5.00 | 0.0074 | 663.01 |
| 5.50 | 0.0089 | 606.39 |
| 6.25 | 0.0084 | 730.09 |
| 7.14 | 0.011 | 636.92 |
| 8.00 | 0.013 | 603.84 |
| 10.00 | 0.0172 | 570.49 |
| 11.76 | 0.0124 | 930.60 |
| 12.50 | 0.0102 | 1202.51 |
| 13.33 | 0.0098 | 1334.70 |



| 14.28 | 0.0115 | 1218.46 |
| 15.00 | 0.0108 | 1362.85 |

**Table 5: $f_{peak}$, $A_{peak}$ and $V_{s30}$ values for 27 stations located in Terai region of Uttarakhand and Delhi**
**region.**

| GINV | | | |
|---|---|---|---|
| Station Code | $f_{peak}$,(Hz) | $A_{peak}$, | $V_{s30}$ (m/s) |
| IGN | 3.6 | 2.6 | 493* |
| JNU | 6.5 | 2.05 | 565* |
| DJB | 5.22 | 3.2 | 543* |
| NDI | 6.8 | 2.5 | 493* |
| IMD | 6 | 2 | 543* |
| NTPC | 2.8 | 3.6 | 345* |
| ANC | 4.6 | 3.2 | 564* |
| JAMI | 4.7 | 3.15 | 346* |
| LDR | 0.7 | 4.32 | 270* |
| VCD | 5.6 | 2.7 | 550* |
| IIT | 4.3 | 2.94 | 332* |
| NSIT | 2.4 | 2.5 | 391* |
| RGD | 2.3 | 2.07 | 346* |
| DLU | 1.81 | 3.2 | 323* |
| DCE | 3.78 | 3.1 | 328* |
| IGI | 2.2 | 2.4 | 360* |
| ZAKI | 3 | 3.49 | 337* |
| ROI | 2.3 | 3.24 | 303* |
| ALIP | 1.4 | 3.09 | 338* |
| DUN | 2.9 | 5.8 | 289** |
| KHA | 1.3 | 4.2 | 218** |
| KSK | 3.13 | 3.75 | 208** |
| RIS | 3.8 | 3 | 331** |
| ROO | 1.16 | 4.35 | 218** |
| TAN | 5.4 | 3.4 | 434** |
| UDH | 2.74 | 6.9 | 198** |
| VIK | 2.29 | 3.8 | 424** |

**Pandey et al., 2016a; * Pandey et al., 2016b













**List of Figures**
1. Figure 1: Map of the region under study with EQs (stars), recording stations (triangles), and paths (solid-
lines).
2. Figure 2: Distribution of hypocentral distances in the data set.
3. Figure 3: S wave spectral attenuation versus hypocentral distance. Note that Log A(f,R$_0$) at reference distance
is zero.
4. Figure 4: Frequency dependence of the quality factor Q for hypocentral distances between 15km to 105km
5. Figure 5: Comparison of Q$_S$ values of North West Himalaya with those obtained from parts of North West
Himalaya and Delhi region. The compared relations for Qs versus frequency are as follows: Garhwal-Kumouan
Himalaya: $Q_S = 175 * f^{0.833}$ (Mukhopadhyay et al., 2010); Kinnaur Himalaya: $Q_S = 86 * f^{0.96}$ (Kumar et al.,
2014) ; Garhwal Himalaya: $Q_S = 151 * f^{0.84}$ (Negi et al., 2015); Delhi and NCR region: $Q_S = 98 * f^{1.07}$
(Sharma et al., 2015).
6. Figure 6: Comparison of Q$_S$ values of this study with regions of different tectonic settings of the world.
7. Figure 7: Site amplification curves obtained using GINV for horizontal component and vertical component
8. Figure 8: Horizontal to vertical ratio curve obtained using GINV and HVSR method
9. Figure 9: Spatial distribution of estimated f$_{peak}$ for: (A) Delhi, (B) Haryana, (C) Himachal Pradesh, (D) Punjab
and (E) Uttarakhand
10. Figure 10: Spatial distribution of estimated A$_{peak}$ for: (A) Delhi, (B) Haryana, (C) Himachal Pradesh, (D)
Punjab and (E) Uttarakhand
11. Figure 11: V$_{s30}$ as a function of f$_{peak}$ (a)  and A$_{peak}$ (b) of GINV method for recording stations at Delhi and
Tarai region of Uttarakhand.
**List of Tables**
Table 1: Detail of strong motion recording stations.
Table 2: Details of earthquakes considered for estimation of site parameters in this work
Table 3: List of Earthquakes and the corresponding stations considered for the estimation of path parameter.
Table 4: Resulting parameters of Eq. (7).
Table 5: f$_{peak}$, A$_{peak}$ and V$_{s30}$ values for 27 stations located in Terai region of Uttarakhand and Delhi region.