# Peer review of "Estimation of path attenuation and site characteristics in the north-west Himalaya and its adjoining area using generalized inversion method"

_Natural Hazards and Earth System Sciences, 2018_

## Referee Comment (RC1) · Anonymous Referee #1 · 17 Sep 2018

General comment:

============

The paper addresses relevant questions concerning the evaluation of the seismic hazard in the area that extends approximately from Delhi to the north-western tranche of Himalaya in North India. The authors focus on two arguments: 1) the attenuation of seismic waves travelling through the Earth's crust in that area (i.e. the "path attenuation") and 2) the spectral amplification of seismic waves at the sites of several

accelerometric stations deployed there (i.e. site characteristics). Unfortunately the quality of the results presented in the paper is not up to the international standards. The part concerning the evaluation of site characteristics is completely flawed by erroneous assumptions and misconceptions, whereas the part concerning the attenuation lacks the estimation of the uncertainty in the results. Moreover the authors neglect possible considerations that may arise from the points of view of geology and seismotectonics and fail to insert appropriately their work in the framework of seismic hazard studies. As a consequence, the presented results are insufficient to support any significant interpretation or conclusion and the contribution of the work to the current state of the art is almost irrelevant. The English language is poor in many critical points and the manuscript appears to be compiled with insufficient attention for potential readers. In consideration of the numerous inconsistencies, I recommend the rejection of the manuscript and suggest the authors a more critical approach in the elaboration of seismological data.

Specific comments:

============

Title:

Considering that the article is submitted to a journal with a wide and diversified audience the title is inappropriate, since it does not mention seismicity or seismic hazard (neither in the keywords!). Apart from this, the title is also inaccurate, referring to NW Himalaya and "adjoining regions", but in the paper only regions belonging to states of India are taken into account.

Abstract:

The abstract does not illustrate the motivations of the work and does not evidence the meaning of the obtained results. Line 2: what is PESMOS? Lines 4-5: was the study performed to demonstrate the presence of the Moho discontinuity in the region? If so,

why do not the authors explicitly declare that (in the title and in the introduction)? If not, why is the presence of Moho discontinuity presented as a foremost result in the abstract? The results presented in the paper are not analyzed with sufficient rigor to support such a claim, anyway. Lines 9-13: sincerely, I do not understand the meaning of these sentences. Lines 13-14: the cited maps have no scientific basis (see later).

Introduction:

Apparently the authors have forgotten that the seismological hazard of an area is strictly linked to geological features. In fact they do not provide any overview of the geological and seismological phenomena that may represent an issue for the seismic hazard evaluation in the studied area. For instance, we know that the Himalayan area is characterized by a peculiar seismo-tectonic regime, which produces earthquakes in a wide depth range and I would expect the authors to discuss how this feature affects the evaluation of the propagation term. And what about the role of the alluvial deposits of the Ganges basin on the seismograms recorded in the stations located in the southern part of the area? Considerations like these would allow the reader to understand better the factors that affect the seismic hazard of the studied area and which methods are most suitable to quantify them. But in the present paper the reader can find only trivial lists of "earthquake parameters" to be inverted with "some spectral modeling" (line 51) and a tedious list of previous works. The importance of the data availability and data quality is also neglected. Line 41-42: what is IS 1893:2002 ? Line 54: again, what is PESMOS? Authors should provide explanation for acronyms and some references would be appreciated, too. Line 61-64: the authors claim that lack of knowledge about geology beneath the recording station does not allow them to identify a reference site, however later in the paper (lines 347 and following, on page 12) they cite the articles of Pandey et al. 2016 (missing in bibliography), which should contain a good analysis of the site characteristics of 27 stations of those used in the present work. Why did non the authors exploit that data to constrain their inversion? That would certainly gave them more reliable results than their "non-reference generalized inversion approach"

(line 63), based on the minimum norm criterion, to be discussed later in this review.

Database:

The authors declare that in the estimation of site characteristics they considered 341 records with corresponding hypocentral distances ranging from 10 to 85 km (line 98), but for the path attenuation they considered only 207 records out of 341, line106), with hypocentral distance ranging from 9 to 200 km (line 109). How can a subset of records be defined over a wider distance range than the initial set? No discussion is given about the signal to noise ratio of the accelerometric data considered in the inversion. Considering the involved distances (up to 200 km) and magnitudes (down to $MW=2.3$) the quality of the records must be verified. No description is given of the enviroment where the stations are located. Apparently no correction for the instrumental response was performed. No references for the hypocentral coordinates are given and PESMOS is again not adequately cited.

Path attenuation:

The inversion for the path attenuation was performed following word by word the approach proposed by Castro et al. (1990). There is no estimation of the uncertainties and the lack of this information deprives the results of their scientific relevance! The authors describe a kink in the attenuation curve which is the only interesting result of their investigation, from the perspective of the application to seismic hazard studies as well as from a purely academical research perspective. Unfortunately the authors do not deepen the analysis of the observed feature and they get rid of the question with a hurried explanation concerning "reflected or refracted waves from the Moho" (line 173) that are observed also elsewhere in the World by other authors. By doing this, the authors miss an opportunity to characterize quantitatively a path attenuation feature which is specific of the studied area and which could have a possible impact on the seismic hazard in this part of India. Instead, they produce a limited range (up to 105 km) parametric attenuation curve which, as they recognize, "falls in between existing

attenuation curves" (line 216), obtained by other studies for more limited regions inside the studied area. This result is indeed plausible, but it does not represent an advancement in the state of the art! Line 144: The character "omega" is usually reserved for the quantity "circular frequency", please use "w" to denote the "weight" terms (Castro et al. (1990)), that are intended here. Lines 154-155: The authors claim they have selected the "length" ("width" I guess) of the bins in a way "such that there is almost equal number of data points in every bin", but according to the histogram in figure 2, where the number of records in each distance bin ranges from zero (!) to 28, I believe the selection could have been done better.

Site effects:

The chapter on the site effects is even less scientifically accurate. Instead of following an established approach (e.g. Hartzell (1992)) the authors attempt a site by site inversion and further interpretations which are conceptually flawed in almost all points. The first mistake consists in the site by site inversion described in the paper (Lines 253-257), which implies different solutions for the same source term involved in wavefroms recorded at different stations (whereas the "sharing" of the source contribution in the waveforms recorded at different sites for the same event represents the "strength" of the approach used by Hartzell (1992)). A second mistake lies in the adopted "minimum norm solution", which has no proper justification: are we perhaps searching amplification functions with minimum norm? A third mistake consists in the introduction of the "site amplification factor" defined as the ratio of the amplification functions inverted from the horizontal and the vertical components (line 300), which is by construction a proxy for the already computed HVSR (I won't spend time in demonstrations here, but it is not difficult to prove it). There is no surprise that the authors found a 1:1 matching in peak frequencies obtained with the two methods (line 307). We can find another big mistake in the part where the parameter Vs30 is evaluated from the relation fpeak=Vz/4H with H taken as 30 m (line 324), which can be understood as a total misinterpretation of the meaning of the relation between fpeak and H. One more mistake

consists in the interpolation of the fpeak (and Apeak) values obtained at each station site in a continuous distribution over the territories of some Indian states (lines 330-345 and figures 9 and 10). The site characteristic has in fact strictly local validity and it is unevenly distributed over the territory, following in principle the geological and morphological features. Whatever extrapolation method was used (the authors do not inform us about that, but no geology seems involved), it has no scientific basis if applied to fpeak and apeak values collected in a restricted set of stations that are located tens or hundreds km apart. The lack of significance of these maps is evident from a mistake in the mistake: the distributions are discontinuous across the borders, as if the site characteristics were dependent on the political borders! The chapter ends with a futile and unmotivated attempt to find an empirical law relating the inverted f-peak and Apeak values and the effectively measured Vs30 obtained by other authors (lines 346-363), an attempt which appears completely detached with the rest of the article. Line 252: equation 12 represents an excessively complicated system. In fact, the problem can be solved for each frequency separately (as it was done for attenuation). Figures 6a-6f: The y-axis is labeled as PSA (pseudo acceleration?) in units of g whereas the legend indicates spectral amplification obtained with GINV method. The values are however inappropriate for spectral amplification (they must be ranging around the value 1). The tick-marks on the X (logarithmic) axis are only two without any grid, thus making the plot nearly useless!

Conclusions:

The conclusions provide an incomplete summary of the manuscript contents without any interesting discussion or meaningful conclusion. The authors claim the "presence of reflected and rifracted arrival from the Moho" (line 369) without any quantitative argumentation and without any discussion about the significance of this effect in respect of the seismic hazard.

---

## Referee Comment (RC2) · Anonymous Referee #2 · 1 Oct 2018

The manuscript "Estimation of path attenuation and site characteristics in the north-west Himalaya and its adjoining area using generalized inversion method" by Hareeshkumar and Kumar is a study concerning the estimate of attenuation and seismic site response considering three components accelerogram recordings. The topic is interesting since the studied area is considered one of the most hazardous in the world for seismicity. However, major revisions are necessary before the publication of the submitted material. In particular, what is new compared to other studies on the same topic? I think that more effort should be done in description of results. Some

comments: The manuscript contains several type mistakes and some sentences are unclear. Therefore, during the revision phase, the authors should pay attention to correct these grammatical errors.

What is the meaning of PESMOS acronym?

In the introduction the authors refer to several geographic places, but no map is shown in the text to help an international reader. Moreover, the authors underline the high level of seismic hazard of the region, but no tectonic setting is described in the text. Probably, an overview of the geologic setting of the area could help the reader.

As concern the recordings, did the authors used some criteria to check the quality of the traces (e.g. signal-to-noise)?

In the Methodology section, there is a considerable amount of extraneous material regarding the theory of the adopted procedures to process the data. These sentences are not central to the results of the paper. Therefore, Some formulae and matrix could be deleted or moved in an appendix.

As concerning Figure 3 more details should be given about the "kink". This result seems to be interesting. Is at the same frequency observed by other authors? What is the Moho depth? Etc. . . Try to better explain.

In site response analysis the authors describe classical HVSR based on Fourier spectra, but starting from line 280 they introduced the ratio of response spectra. In this case it is important to describe the differences. H/V in Fourier domain are different from H/V in response spectra.

Figure 7 and 8 should be arranged in a different way.

The authors should explain why 6.75 Hz is used to discriminate soil and rock sites. In the paper D'Alessandro et al. (2012) there is a classification of the H/V as a function of peak.

The method adopted to relate the frequency and depth with Vs,30 is less clear and speculative. For a frequency the bedrock could be at different depth as a function of thickness and velocity.

Geographic distribution of amplitudes and frequencies of the spectral ratios are not scientifically relevant considering the dimensions of the studied area. Probably distribution charts of frequencies and amplitudes observed at the investigated stations could be more interesting to subdivide these.

Check the reference list, is incomplete (e.g. Alessandro et al. 2012, is D'Alessandro et al. 2012?).
* * *

---

## Author Comment (AC1) · 23 Nov 2018

General Comment: The paper addresses relevant questions concerning the evaluation of the seismic hazard in the area that extends approximately from Delhi to the north-western tranche of Himalaya in North India. The authors focus on two arguments: 1) the attenuation of seismic waves travelling through the Earth's crust in that area (i.e. the "path attenuation") and 2) the spectral amplification of seismic waves at the sites of several accelerometric stations deployed there (i.e. site characteristics). Unfortunately the quality of the results presented in the paper is not up to the international

standards. The English language is poor in many critical points and the manuscript appears to be compiled with insufficient attention for potential readers. In consideration of the numerous inconsistencies, I recommend the rejection of the manuscript and suggest the authors a more critical approach in the elaboration of seismological data. Response: The authors thank the reviewer for the valuable time in reviewing the work and giving expert comments. Going with first observation of the reviewer as highlighted above (The paper addresses relevant questions concerning the evaluation of the seismic hazard in the area that extends approximately from Delhi to the north-western tranche of Himalaya in North India), authors want to highlight here that specifically this paper does not attempt at all the seismic hazard of the study area. Rather the objective of present work is to quantify the path attenuation and site characteristics of the study area. Though detailed seismic hazard requires the knowledge of path attenuation and site characteristics, but other informations are also required while attempting seismic hazard of an area, which are neither attempted nor claimed in this work. Authors nowhere conclude anything on seismic hazard of the study area based on present findings. Further, authors want to highlight that regional information on path attenuation as well as site characteristics, for the present study area are still missing and thus are attempted in this work. For this reason, either for site specific ground response analysis or for ground motion simulation, still researchers are practising utilization of above parameters from other regions. Authors believe that unless above parameters are estimated on regional scale, end results will have no to limited application in understanding ongoing seismicity mapping of the study area. English of the manuscript has been improved throughout the manuscript while revising. Comment: The part concerning the evaluation of site characteristics is completely flawed by erroneous assumptions and misconceptions, whereas the part concerning the attenuation lacks the estimation of the uncertainty in the results. Moreover the authors neglect possible considerations that may arise from the points of view of geology and Seismotectonic and fail to insert appropriately their work in the framework of seismic hazard studies. As a consequence, the presented results are insufficient to support

any significant interpretation or conclusion and the contribution of the work to the current state of the art is almost irrelevant. Response: Authors want to highlight that in the current paper site effects are characterised based on the predominant frequency obtained from HVSR and non-reference Generalized inversion technique (GINV). The effectiveness of both HVSR and GINV with regards to estimation of site characteristics is well documented in literatures (For eg. Field and Jacob, 1995; Harinarayan and Kumar 2017b). Further, the reviewer commented that the outcome of the present study does not give possible considerations that may arise from the points of view of geology and Seismotectonic. Authors appreciate the reviewers comment and agree that incorporating geological and Seismotectonic consideration would be good. However, authors want to emphasise here that numerous published literature highlights that PESMOS recording stations are lacking with geological information. The site class given by PESMOS is based on physical description of surface materials, local geology following Seismotectonic Atlas of India (GIS 2000), Geological Maps of Indian and not based on actual field investigation (Kumar et al., 2012). Geophysical subsurface exploration studies on some of the recording stations in northwest Himalaya reported by Pandey et al., 2016a; b have highlighted the flaws in the site class given by PESMOS, where recording stations classified as on rock site were found to be on soil sites. The lack of accurate information with regards to geology for the recording stations maintained by PESMOS prevent the scope for any studies from the point of view of geology. Authors would like to highlight here that it is one among many reasons which motivated the authors to go for site characterization of the PESMOS recording station of the study area, as done in this work. The above discussion has been added in Line 50-64 in the revised manuscript. In addition, present paper estimates path attenuation and site characteristics based on earthquake records. There is no discussion made with regards to source parameters in the manuscript. Hence, authors feel that incorporating the Seismotectonic of the study area is not relevant referring to present findings and are beyond the scope of the present study. Further, the reviewer suggested to incorporate the outcomes of the present work in the frame work of seismic hazard studies. As

highlighted in earlier comment that present findings are not sufficient alone for seismic hazard quantification and thus presenting the presenting findings in terms of seismic hazard analysis is not possible and is beyond the scope of this work. Further, with limited and partial information available from present study, if seismic hazard is attempted, authors believe that such seismic hazard will not be appropriate and does not convey anything related to actual seismicity of the region. However, considering reviewer's suggestion, possible comment on seismicity of the region, in a broader sense, based on the quality factor obtained in the present study, is made in Line 242-244 in the revised manuscript. Comment 1: Title: Considering that the article is submitted to a journal with a wide and diversified audience the title is inappropriate, since it does not mention seismicity or seismic hazard (neither in the keywords!). Response: As per reviewer's suggestion, keywords have been modified incorporating the term "seismicity". Comment 2: Apart from this, the title is also inaccurate, referring to NW Himalaya and "adjoining regions", but in the paper only regions belonging to states of India are taken into account. Response: As per reviewer's suggestion, the title of the paper has been modified to: "Estimation of path attenuation and site characteristics in the northwest Himalaya and its adjoining area within India territory, using generalized inversion method". Comment 3: Line 2: what is PESMOS? Response: PESMOS stands for Program for Excellence in Strong Motion Studies. PESMOS maintains ground motion records from recording stations installed in various regions within India by the Government of India to monitor the ongoing seismicity. Earthquake records since 2004 are available in PESMOS data base. A brief description of PESOM database has been added in Line 49 – 60 in the revised manuscript. Comment 4: Lines 4-5: was the study performed to demonstrate the presence of the Moho discontinuity in the region? If so, why do not the authors explicitly declare that (in the title and in the introduction)? If not, why is the presence of Moho discontinuity presented as a foremost result in the abstract? Response: Authors want to highlight here that the foremost objective of this paper is to estimate path attenuation and site characteristics based on strong motion records which has been clearly mentioned in the title as well. Identifying the presence

of Moho discontinuity based on the nature of the attenuation curve is an additional finding of the work. Authors believe that though this is an important finding, still it is based on estimation of path attenuation and hence should be mentioned in the abstract and not to be included in the title. Comment 5: Lines 9-13: sincerely, I do not understand the meaning of these sentences. Response: The line 9-13 from the original manuscript has been rewritten as follows; "The ratio of the horizontal and vertical site amplification components is computed to determine the amplification function and predominant frequency for each of the recording stations. The amplification function and predominant frequency based on generalized inversion method is compared with that obtained from horizontal to vertical spectral ratio (of S wave portion of the accelerogram) method." in Line 9-13 in the revised manuscript.

Comment 6: Lines 13-14: the cited maps have no scientific basis. Response: As per reviewer's suggestion, discussion on spatial distribution of predominant frequencies and amplification functions has been removed in the revised manuscript. Comment 6: Introduction: Apparently the authors have forgotten that the seismological hazard of an area is strictly linked to geological features. In fact they do not provide any overview of the geological and seismological phenomena that may represent an issue for the seismic hazard evaluation in the studied area. Response: The Authors thank the reviewer for this suggestion. As highlighted earlier, information on geology of the recording station is not available. However, an overview of Seismotectonic settings of the study has been added in line 95-113 in the revised manuscript. Comment 7: And what about the role of the alluvial deposits of the Ganges basin on the seismograms recorded in the stations located in the southern part of the area? Considerations like these would allow the reader to understand better the factors that affect the seismic hazard of the studied area and which methods are most suitable to quantify them. Response: Authors completely agree that the characteristics of alluvial deposits in the Ganga basin will change the seismogram characteristics. In case such characteristics are known based on suitable in-situ investigation, it can be studied further. Since present work attempt to understand subsoil characteristics completely based on ground motion records in terms

of fpeak and Apeak, incorporating alluvial characteristics is beyond present work's objective. Possible change in seismogram if any will also be reflected in fpeak value determined in this work. Collectively, neither data required for such study is not available at present nor it will affect the present findings of fpeak. Comment 8: Line 41-42: what is IS 1893:2002 ? Response: IS 1893:2016 is an Indian seismic code. The reference for IS 1893:2016 is: IS 1893: Part 1–2016. Indian standard criteria for earthquake resistant design of structures—part 1: General Provisions and Buildings, Bureau of Indian Standards, New Delhi, India. Comment 9: Line 61-64: the authors claim that lack of knowledge about geology beneath the recording station does not allow them to identify a reference site, however later in the paper (lines 347 and following, on page 12) they cite the articles of Pandey et al. 2016 (missing in bibliography), which should contain a good analysis of the site characteristics of 27 stations of those used in the present work. Why did non the authors exploit that data to constrain their inversion? That would certainly gave them more reliable results than their "non-reference generalized inversion approach" (line 63), based on the minimum norm criterion, to be discussed later in this review. Response: Based on the reviewers comment on "lack of knowledge about geology beneath the recording station", the authors want to highlight that the PESMOS database lacks accurate information of subsurface for majority of recording stations. The site class given by PESMOS is based on physical description of surface materials, local geology following Seismotectonic Atlas of India (GIS 2000), Geological Maps of Indian and not based on actual field investigation (Kumar et al., 2012). Geophysical subsurface exploration studies on some of the recording stations in northwest Himalaya reported by Pandey et al., 2016a; b have highlighted the flaws in the site class given by PESMOS, where recording stations classified as on rock site were found to be on soil sites. The above discussions have been incorporated in line 55-64 in the revised manuscript. Authors also want to highlight here that in order to perform conventional generalized inversion method (Andrews, 1986; Hartzell, 1992), reference site (usually rock site) is required in-order to remove the trade-off between the source and site parameters. However, the field study by Pandey et al., 2016a; b

used 27 stations and which are actually soil sites. Thus, it is not possible to identify reference site (rock sites) based on the findings of Pandey et al., 2016a; b. For this reason, authors feel non-reference generalized inversion approach is a better alternative to estimate the site term in this work. Based on above discussion, the referred statements are corrected accordingly in the revised manuscript. Comment 10: The authors declare that in the estimation of site characteristics they considered 341 records with corresponding hypocentral distances ranging from 10 to 85 km (line 98), but for the path attenuation they considered only 207 records out of 341, line106), with hypocentral distance ranging from 9 to 200 km (line 109). How can a subset of records be defined over a wider distance range than the initial set? Response: The authors are thankful to the reviewer comment. Following correction have been made in the above referred statement; "For estimating site characteristics, 341 records from 86 EQs, with magnitudes ranging from Mw=2.3 to Mw=5.8, having focal depths ranging from 2 to 80km are used (Line 121 in the revised manuscript). The database for estimating path attenuation consists of 207 records from 32 EQs, with magnitude ranging from Mw= 3.1 to Mw=5.5, focal depths from 3 to 55km, hypocentral distance from 9 to 200km" in line 130 in the revised manuscript. Comment 11: No discussion is given about the signal to noise ratio of the accelerometric data considered in the inversion. Considering the involved distances (up to 200 km) and magnitudes (down to MW=2.3) the quality of the records must be verified. No description is given of the environment where the stations are located. Response: Authors want to highlight that signal to pre-event noise (all of equal window length) ratio (SNR) for all the records were computed and records with SNR greater than 5 similar to (Ameri et al., 2011) are considered for the present analysis. The discussion on SNR has been incorporated in Line 136 in the revised manuscript. Comment 12: No references for the hypocentral coordinates are given and PESMOS is again not adequately cited. Response: As per the reviewer's suggestion, references for the hypocentral coordinates have been added in the revised manuscript. Authors want highlight here that the information regarding the coordinates of earthquake epicentre and recording stations are obtained from PESMOS. Comment

13: There is no estimation of the uncertainties and the lack of this information deprives the results of their scientific relevance! Response: Authors agree with the reviewer's observation. As per the reviewer's suggestion, the uncertainties in the quality factor estimation has been added in the line 227 in the revised manuscript. Comment 14: The authors describe a kink in the attenuation curve which is the only interesting result of their investigation, from the perspective of the application to seismic hazard studies as well as from a purely academic research perspective. Unfortunately the authors do not deepen the analysis of the observed feature and they get rid of the question with a hurried explanation concerning "reflected or refracted waves from the Moho" (line 173) that are observed also elsewhere in the World by other authors. By doing this, the authors miss an opportunity to characterize quantitatively a path attenuation feature which is specific of the studied area and which could have a possible impact on the seismic hazard in this part of India. Response: Authors want to emphasis here that the scope of this paper is to estimate path attenuation and site characteristics based on strong motion records. Identifying the presence of Moho discontinuity based on the nature of attenuation curve is an additional finding of the work. Any further information regarding Moho discontinuity cannot be deducted from the present study. Comment 15: Line 144: The character "omega" is usually reserved for the quantity "circular frequency", please use "w" to denote the "weight" terms (Castro et al. (1990)), that are intended here. Response: As per the reviewer's suggestion, weighing factors have been denoted by w in the revised manuscript. Comment 16: Lines 154-155: The authors claim they have selected the "length" ("width" I guess) of the bins in a way "such that there is almost equal number of data points in every bin", but according to the histogram in figure 2, where the number of records in each distance bin ranges from zero (!) to 28, I believe the selection could have been done better. Response: Authors agree with the reviewer's suggestion regarding replacing the term "length" in Lines 154-155 to "width". As per the reviewer's suggestion the term "length" has been replaced to "width" in Line 547 in the revised manuscript. Authors want to highlight here that the width of the bins are selected such that there is sufficient number of data points in

every bin. Moreover, it can be observed from Figure 2 that there are only 2 records at 125km bin, 0 records in 135km bin, 1 record in 145km bin, and 3 records in 155km bin. Collectively there very few records in bins beyond 115km. For this reason, EQ records with hypocentral distance only up to 115km are considered for the analysis. Comment 17: The chapter on the site effects is even less scientifically accurate. Instead of following an established approach (e.g. Hartzell (1992)) the authors attempt a site by site inversion and further interpretations which are conceptually flawed in almost all points. Response: The reviewer commented on the use of non-reference GINV over the established approach (reference GINV), which requires earthquake data recorded on rock site. The authors have earlier highlighted the problem in identifying recording stations on rock sites. Authors want to highlight here that performing inversion similar to Hartzell (1992) in this context is not feasible. Authors feel modifying the generalized inversion method such that analysis can be carried out without reference site condition is a better alternative as done in this work. Comment 18: The first mistake consists in the site by site inversion described in the paper (Lines 253- 257), which implies different solutions for the same source term involved in wavefroms recorded at different stations (whereas the "sharing" of the source contribution in the waveforms recorded at different sites for the same event represents the "strength" of the approach used by Hartzell (1992)) Response: Authors partially agree with the reviewer's remark about the "sharing" of the source contribution in the waveforms recorded at different sites for the same event represents the "strength" of the approach used by Hartzell (1992). Authors want to highlight here that the objective of the approach used by Hartzell (1992) is to simultaneously determine the source and site component. Authors want to emphasis here that the objective of the non-reference GINV is to determine the site component alone and no source component. Further, authors nowhere claim to determine source component in the manuscript. Hence, obtaining different solutions for the same source term involved in waveforms recorded at different stations will not affect the obtained value of site component, as obtained in this work. Comment 19: A second mistake lies in the adopted "minimum norm solution", which has no proper justification: are we perhaps searching amplification functions with minimum norm? Response: Authors want to highlight that minimum norm inversion procedure (also known as Moore- Penrose matrix inversion) used to estimate soil term is in accordance with Penrose, (1955). A brief description regarding the inversion procedure is given below. Eq. 11 (in the revised manuscript) represents a linear system of the form: Ax = B, where B is the data vector containingãĂŰ dãĂŮ_ijj, x is the vector containing the model parameters [g(f) and s_i (f)], and A is the system matrix relating x and B as described in eq. 12 (Menke, 1989). ãĂŰ ãĂŰlnUãĂŮˆA (f)ãĂŮ_(ij )= ãĂŰln S(f)ãĂŮ_i+ ãĂŰln G(f)ãĂŮ_j (11) ← A → ←x→ ←B→ ← 1st event → ← nth event → ← Site effect→ 1 2 ... m 1 2 ... m 1 2 ... m 1 0 ... 0 ...... 0 0 0 1 0 ... 0 s1(f1) d1(f1) 0 1 0 ...... 0 0 0 0 1 ... 0 : : : : : : : : : : : : : : : : : : : : : : 0 0 0 1 0 0 ... 0 0 0 ... 1 s1(fn) d1(fm)

sn(f1) = : For nth earthquake sn(fn) dn(fm) 0 0 ... 0 ...... 1 ... 0 1 0 ... 0 g(f1) dn(fm) 0 0 ... 0 ...... 0 1 ... 0 0 1 ... 0 g(f2) : : : : : : : : : : : : : : : : : : : : : : : : : : 0 0 ... 0 ... 0 0 ... 1 0 0 ... 1 g(fm) dn(fm) (12) The matrix form in Eq. (12) represents a purely under determinate system since there are (n+1)×m parameters for 'm × n' data (here m is the number of sample frequency and n is the number of EQs recorded at a particular recording station). Further, the model parameters at the selected frequencies are obtained using Moore- Penrose matrix inversion (or Minimum norm inversion) method given Penrose, (1955) as given below. x=(AˆH.A)ˆ(-1).AˆH.B In the above equation, AˆH is the conjugate transpose of matrix A . Comment 20: A third mistake consists in the introduction of the "site amplification factor" defined as the ratio of the amplification functions inverted from the horizontal and the vertical components (line 300), which is by construction a proxy for the already computed HVSR (I won't spend time in demonstrations here, but it is not difficult to prove it). There is no surprise that the authors found a 1:1 matching in peak frequencies obtained with the two methods (line 307). Response: Authors want to highlight here that in the present study site amplification curve for each recording station is calculated based on two different methods i.e., HVSR and non-reference GINV. Effectiveness of HVSR method in providing good estimate of predominant frequency (fpeak) is well documented (for eg.

Field and Jacob, 1995; Zhao et al., 2006). Authors feel that similarities in the value of fpeak obtained using non-reference GINV with that obtained using HVSR show the robustness of the non-reference GINV used in the present study. Comment 21: We can find another big mistake in the part where the parameter Vs30 is evaluated from the relation fpeak=Vz/4H with H taken as 30 m (line 324), which can be understood as a total misinterpretation of the meaning of the relation between fpeak and H. Response: The relation fpeak=Vz/4H given by Kramer (1995) is used to estimate the range of fpeak values corresponding to the range of Vs30 for site class as per NEHRP classification scheme. Similar to the present work Zhao et al. (2006) carried out site classification of 874 recording stations in Japan based on HVSR method. Zhao et al. (2006) gave possible range of fpeak for rock sites as greater than 5Hz and soil sites less than 5Hz respectively. Further, the range of fpeak obtained using the relation given by Kramer, (1996) in the present study is: fpeak $\geq$ 6.35Hz for rock site and fpeak< 6.35Hz for soil sites. The possible range of fpeak for rock sites and soil sites given by Zhao et al. (2006), are closely matching with fpeak range obtained based on the equation fpeak=Vz/4H.

Comment 22: One more mistake consists in the interpolation of the fpeak (and Apeak) values obtained at each station site in a continuous distribution over the territories of some Indian states (lines 330-345 and figures 9 and 10). The site characteristic has in fact strictly local validity and it is unevenly distributed over the territory, following in principle the geological and morphological features. Whatever extrapolation method was used (the authors do not inform us about that, but no geology seems involved), it has no scientific basis if applied to fpeak and Apeak values collected in a restricted set of stations that are located tens or hundreds km apart. The lack of significance of these maps is evident from a mistake in the mistake: the distributions are discontinuous across the borders, as if the site characteristics were dependent on the political borders! Response: Authors agree with the reviewer's observation. As per reviewer's suggestion, discussion on spatial distribution of predominant frequencies and amplification functions has been removed from the revised manuscript. Comment 23: Figures

6a-6f: The y-axis is labeled as PSA (pseudo acceleration?) in units of g whereas the legend indicates spectral amplification obtained with GINV method. The values are however inappropriate for spectral amplification (they must be ranging around the value 1). The tick-marks on the X (logarithmic) axis are only two without any grid, thus making the plot nearly useless! Response: The y axis represents spectral amplitude (acceleration). Figures 6a-6f have been modified considering the reviewers suggestions. Comment 24: Conclusions: The conclusions provide an incomplete summary of the manuscript contents without any interesting discussion or meaningful conclusion. The authors claim the "presence of reflected and refracted arrival from the Moho" (line 369) without any quantitative argumentation and without any discussion about the significance of this effect in respect of the seismic hazard. Response: Authors would like to highlight here that observation about Moho in the present study is purely based on the kink observed in attenuation curves in this work and observing similar trend by earlier researchers. Further comment on this observation required detailed studies and is beyond the scope of this work. While revising the manuscript, "Conclusion" section is completely rewritten clearly citing the important observations from the present work. In addition, need for detailed investigation for Moho discontinuity, which is beyond the scope of present work, is also highlighted clearly in the conclusion.

Please also note the supplement to this comment:
https://www.nat-hazards-earth-syst-sci-discuss.net/nhess-2018-148/nhess-2018-148-AC1-supplement.pdf

---

## Author Comment (AC2) · 23 Nov 2018

Response to referee 2: The manuscript "Estimation of path attenuation and site characteristics in the northwest Himalaya and its adjoining area using generalized inversion method" by Hareeshkumar and Kumar is a study concerning the estimate of attenuation and seismic site response considering three components accelerogram recordings. The topic is interesting since the studied area is considered one of the most hazardous in the world for seismicity. However, major revisions are necessary before the publication of the submitted material. In particular, what is new compared to

other studies on the same topic? I think that more effort should be done in description of results. Some comments: The manuscript contains several type mistakes and some sentences are unclear. Therefore, during the revision phase, the authors should pay attention to correct these grammatical errors. Response: The authors express their extended gratitude to the reviewers for reviewing the manuscript and giving their useful comments. Detailed descriptions explaining the findings form present work have been added throughout the manuscript. In addition, the entire manuscript has been checked for any possible mistake or any kind and corrected. Further, all the comments of the author have been addressed in the revised manuscript in blue colour. Comment 1: What is the meaning of PESMOS acronym? Response: PESMOS stands for "Program for Excellence in Strong Motion Studies". PESMOS maintains ground motion records from recording stations installed in various regions within India, by the Government of India, to monitor the ongoing seismicity. Earthquake records since 2004 are available in PESMOS database. A brief description of PESMOS database has been added in Line 50 – 64 in the revised manuscript. Comment 2: In the introduction the authors refer to several geographic places, but no map is shown in the text to help an international reader. Response: As per reviewer's suggestion, map incorporating details mentioned in the introduction section related to geographic details has been added (Figure 1) in the revised manuscript. Comment 3: Moreover, the authors underline the high level of seismic hazard of the region, but no tectonic setting is described in the text. Probably, an overview of the geologic setting of the area could help the reader. Response: As per reviewer's suggestion, an overview of the geologic setting of the area has been added in Line 136-137 in the revised manuscript Comment 4: As concern the recordings, did the authors used some criteria to check the quality of the traces (e.g. signal-to-noise)? Response: Authors want to highlight that signal to pre-event noise (all of equal window length) ratio (SNR) for all the records were computed and records with SNR greater than 5(similar to work by Ameri et al., 2011) are considered for analysis. Needful discussion on SNR has been incorporated in Line 136-137 in the revised manuscript. Comment 5: In the

[Figure]

Methodology section, there is a considerable amount of extraneous material regarding the theory of the adopted procedures to process the data. These sentences are not central to the results of the paper. Therefore, some formulae and matrix could be deleted or moved in an appendix. Response: As per reviewer's suggestion some formulae and matrices in Methodology section has been moved to Appendix as can be observed in the revised manuscript. Comment 6: As concerning Figure 3 more details should be given about the "kink". This result seems to be interesting. Is at the same frequency observed by other authors? What is the Moho depth? Etc. . . Try to better explain. Response: Bindi et al., (2004) and Oth et al., (2011) observed kinks in the attenuation curves for the Umbria Marche and Japan regions respectively. Bindi et al., (2004) observed a kink in attenuation curves for frequencies less than 2.24Hz, beyond 40km hypocentral distance. Similarly, Oth et al., (2011) observed a kink in attenuation curves for frequencies less than 2Hz, beyond 90km hypocentral distance. In the present study, a similar kink in the attenuation curves is observed beyond 105km at frequencies less than 5.5Hz. Oth et al., (2011) attributed the above kink in the attenuation curves to the presence of Moho discontinuity in the region. Following Oth et al., (2011), a similar conclusion has been made regarding the kink in the attenuation curves observed in the present study. Studied on crustal imaging of north-west Himalaya by Saikia et al., (2015) also suggests the depth of Moho varying in the range 37km to 52km. Authors want to highlight here that based on the nature of attenuation curves developed in the present study, no conclusion regarding the Moho depth can be made. Comment 7: In site response analysis the authors describe classical HVSR based on Fourier spectra, but starting from line 280 they introduced the ratio of response spectra. In this case it is important to describe the differences. H/V in Fourier domain are different from H/V in response spectra. Response: Authors want to clarify that HVSR calculations in the present study are carried based on response spectra and not based on Fourier spectra. The line 280 ( "Calculate the FAS for the three components [north-south, east-west and vertical] of ground motion records) from original manuscript has been corrected to "Calculate the response spectra considering

5% damping for the three components (north-south, east-west and vertical) of ground motion records" in Line 302 in the revised manuscript. .Comment 8: The authors should explain why 6.75 Hz is used to discriminate soil and rock sites. In the paper D'Alessandro et al. (2012) there is a classification of the H/V as a function of peak. The method adopted to relate the frequency and depth with Vs30 is less clear and speculative. For a frequency the bedrock could be at different depth as a function of thickness and velocity. Response: Authors want to highlight that one of the objective of the present work is to classify recording station as either rock site or soil site based on the predominant frequency (fpeak) obtained from generalized inversion and HVSR analyses. D'Alessandro et al. (2012) attempted to classify recording stations based on HVSR results. Based on the work, D'Alessandro et al. (2012) gave possible ranges of fpeak for rock sites as greater than 5Hz and for soil sites to be less than 5Hz respectively. Further, the range of fpeak obtained using Eq. (1) (in accordance with Kramer, 1996) in the present is: fpeak $\geq$ 6.35Hz for rock site and fpeak< 6.35Hz for soil sites. Thus, possible ranges of fpeak for rock sites and soil sites given by D'Alessandro et al. (2012), are closely matching with fpeak range obtained based on equation below, as adopted in the present work. f_peak= V_z/4H (1) Comment 9: Geographic distribution of amplitudes and frequencies of the spectral ratios are not scientifically relevant considering the dimensions of the studied area. Probably distribution charts of frequencies and amplitudes observed at the investigated stations could be more interesting to subdivide these. Response: The authors thank the reviewer for this useful comment. As per reviewer's suggestion, discussion on spatial distribution of predominant frequencies and amplification functions and corresponding figures has been removed while revising the manuscript. Comment 10: Check the reference list, is incomplete (e.g. Alessandro et al. 2012, is D'Alessandro et al. 2012?). Response: As per reviewer's suggestion, the above reference has been corrected in Line 293 in the revised manuscript. Further, the reference list is also checked for any other incompleteness.

Please also note the supplement to this comment:
https://www.nat-hazards-earth-syst-sci-discuss.net/nhess-2018-148/nhess-2018-148-AC2-supplement.pdf
* * *

---

## Author Comment (AC3) · 23 Nov 2018

**Estimation of path attenuation and site characteristics in the north-west Himalaya and its adjoining area within Indian Territory using generalized inversion method**

Harinarayan, NH[1], Abhishek Kumar[*2]

[1, 2] Department of Civil Engineering, Indian Institute of Technology Guwahati, Assam, India.

Corresponding to: Kumar Abhishek (abhitoaashu@gmail.com/ abhiak@iitg.ernet.in)

**Abstract.** Present work focuses on the determination of path attenuation and site characteristics of earthquake recording stations, located in the north-west Himalaya and its adjoining region, within India. The work is done using two-step generalized inversion technique. In the first step of inversion, non-parametric attenuation curves are developed. $Q_s = (105 \pm 11)f^{(0.94\pm0.08)}$ as S wave quality factor within 105km, is obtained indicating that the region is possibly heterogeneous as well as seismically active. In addition, presence of a kink is observed at around 105km hypocentral distance while correlating path attenuation with the hypocentral distance, indicating the presence of Moho discontinuity in the region. In the second step of inversion, site amplification curves are developed separately from the attenuation corrected data for horizontal and vertical components of the accelerograms. The amplification function and predominant frequency of each recording station based on generalized inversion method is estimated and compared with that obtained from horizontal to vertical spectral ratio (of S wave portion of the accelerograms) method. Path attenuation and site characteristics obtained in the present study are very essential for developing regional ground motion model and for the seismic hazard assessment of the above study area.

Keywords: *Seismicity, Northwest Himalaya, path attenuation, site characteristics, Generalized Inversion, HVSR*

**1 Introduction**

The Himalayan arc extending approximately 2500km between Kashmir and Arunachal Pradesh is one among the seismically most active regions across the globe. Seismic activity of this region can be understood based on induced damages witnessed primarily during 4 great earthquakes (EQs) including 1897 Shillong EQ, 1905 Kangra EQ, 1950 Assam EQ and 1934 Bihar-Nepal EQ, occurred in the last 120 years. Based on seismic activity, the entire Himalayan belt can be subdivided into three distinct segments namely the western, the central and the eastern Himalayas (Philip, 2014). The region of the north-west Himalaya and its foothills, within India come under seismic zone IV and V as per IS 1893: 2016, indicating regions of high to very high seismicity.

Intensity of ground shaking during an EQ, at a particular site is a collective effect of source, path and site parameters. Source parameters include magnitude, fault mechanism, stress drop and rupture process. On the other hand, path parameters include geometric attenuation and loss of seismic energy due to the anelasticity of the earth and scattering of elastic waves in heterogeneous media. Similarly, site characteristics include modification of amplitude, frequency content and duration of the incoming seismic wave by subsurface medium as reaches the surface. Determinations of aforementioned EQ parameters are important for the development of region specific ground motion models, which can further be used for region/site specific seismic hazard assessment (Baro et al., 2018). Above parameters can be estimated from EQ records based on some spectral modelling or inversion approach like generalized inversion method (Andrews, 1986; Castro et al., 1990; Oth et al., 2009 etc.).

To understand the on-going seismicity of various regions within India, the Government of India has installed number of EQ recording stations. EQ records from these stations since 2004 are maintained by PESMOS (Program for Excellence in Strong Motion Studies), which is currently one of the most significant resource of ground motion records in India. At present, PESMOS manages EQ records from 300 recording stations which are distributed in the northern and northeaster parts of India as well as in the Andaman and Nicobar Islands (Kumar et al. 2012). It must be highlighted here that PESMOS database is lacking in terms of accurate information about subsurface for majority of recording stations (Harinarayan and Kumar, 2018). Site class given by PESMOS is based on physical description of surface materials, local geology following Seismotectonic Atlas of India (GIS 2000) and Geological Maps of Indian (GSI, 1998) and not based on actual field investigation (Kumar et al., 2012). In the absence of accurate information of local soil, utilizing EQ records from PESMOS database for seismic studies is a major challenge. Geophysical subsurface exploration studies on some of the recording stations in northwest Himalaya reported by Pandey et al., (2016a; b) had highlighted the flaws in site class given by PESMOS. There are recording stations classified to be on rock site but were found to be on soil sites by Pandey et al., (2016a; b). Harinarayan and Kumar (2018) also reported problems in the subsurface information given by PESMOS and attempted site classification of PESMOS recording stations

In the present study, EQ records from the region of the north-west Himalaya and its foothills within India, as obtained from PESMOS database, are analysed for estimating path attenuation and site characteristics separately, using a two-step generalized inversion of the S-wave Fourier spectra (hereafter referred to as GINV). In the first step, attenuation curves are developed using a non-parametric inversion approach (similar to Castro et al., 1990 and Oth et al., 2008). In the conventional generalized inversion method (Andrews, 1986; Hartzell, 1992), the second step of inversion calculates both site and source spectra, by inverting the S-wave (or Coda wave) spectra, corrected for the path parameter. This method however requires one or more reference sites (usually rock sites) in-order to remove the trade-off between the source and site parameters (Andrews 1986). In the absence of subsoil information for majority of recording stations managed by PESMOS as highlighted above, identifying reference site is not possible. For this reason, only the site parameters are evaluated in the second step of inversion, using a non-reference generalized inversion approach (similar to the work by Joshi et al., 2010; Harinarayan and Kumar 2017b). Further, Obtained site terms are compared with the one calculated from horizontal to vertical spectral ratios (HVSR) from the same S-wave window as used in the GINV.

This study is one of its kind, which systematically evaluates path and site parameters using a larger and regional database. Existing studies on attenuation characteristics determination for the northwest Himalaya used few EQ records that too from limited recording stations. Joshi (2006) estimated frequency independent S wave quality factor ($Q_s$) for the Garhwal Himalayas using 1991 Uttarkashi EQ and 1999 Chamoli EQ ground motion records from 8 recording stations. In another study, Singh et al., (2012) estimated frequency depended $Q_s$ for the Kumaun Himalayas using 23 EQ events, from 9 recording stations by applying the extended coda-normalization method. Similarly, Negi et al., (2015), Banerjee and Kumar, (2017) and Tripathi et al., (2015) estimated $Q_s$ for the Garhwal Himalayas. The aforementioned studies did not highlight the attenuation characteristics of the entire north-west Himalaya region.

Similar to path attenuation studies, very few studies on the determination of site characteristics from EQ records also exist for this region. Nath et al., (2002) computed site terms using the aftershocks of the 1999 Chamoli EQ, obtained from 5 recording stations located in the Uttarakhand region. Similarly, Sharma et al., (2014) estimated site parameters for the Garhwal region of Uttarakhand using EQ records in context of generalized inversion and HVSR. In another work, Harinarayan and Kumar (2017) reported a comparative study on site characteristics computed using EQ records from Tarai region of Uttarakhand using multiple analytical approaches. In another recent work, Harinarayan and Kumar (2018) computed site parameters for recording stations in the northwest Himalayas in terms of predominant frequency ($f_{peak}$) alone using HVSR method.

**2 Study Area**

Present study area includes states of Himachal Pradesh, Uttarakhand, Punjab, Haryana and national capital city, New Delhi, covering an area between 28º N to 34º N latitude and 75.8º E to 80.5º E longitude. According to 2011 Census, the region has a population of 96 million. From seismicity point of view, major Seismotectonic features of the present study are characterized by three north-dipping thrust systems such as the Central thrust (MCT), the Main Boundary thrust (MBT) and the Himalayan frontal thrust (HFT) (Valdiya, 1981). Other tectonic features includes, the Jhelum Balakot fault, the Drang thrust, the Lesser Himalayan Crystalline Nappes, the Jammu thrust, the Vaikrita thrust, the Karakoram fault, the Jwala Mukhi thrust, and the Ramgarh thrust. Both the MCT and the MBT lie parallel to each other within western Himalayan region and were produced during the Cenozonic shortening (Malik and Nakata 2003). The HFT is the youngest active thrust separating the Himalaya region and the Indo-Gangetic alluvial plain (Kumar et al. 2009). The HFT, the MBT and the MCT have generated major EQs in this region (Philip et al. 2014).

| 107 | Two of the most damage inducing EQs, in the last 120 years, in the west Himalayan regions include |
| 108 | 1905 Kangra-Himachal Pradesh EQ (Ms=7.8) (Ambraseys and Douglas 2004) and 2005 Muzzafarbad-Kashmir |
| 109 | EQ (Mw=7.6) (Avouac et al. 2006). Both of these EQs caused severe loss to life and property. Recent EQs of |
| 110 | 1991 Uttarkashi (Mw=6.8) and 1999 Chamoli (Mw=6.6) had occurred on the MCT zone (Harbindu et al. 2014). |
| 111 | The 1999 Chamoli EQ caused a huge landslide in Gopeshwar situated less than 2km northwest of Chamoli city |
| 112 | (Sarkar et al. 2001). This EQ also caused shaking in Chandigarh and Delhi, located far away from the epicentre |
| 113 | (Mundepi et al., 2010). |

**3 Database**

Ground motion records used in this study consists of three components accelerograms obtained from PESMOS database available at http://www.pesmos.in/. The instrumentation used for recording EQs consists of internal AC-63 GeoSIG triaxial force balanced accelerometers and GSR-18 GeoSIG 18 bit digitizers with external GPS (Kumar et al., 2012). Further, ground motion recordings are done in trigger mode during each EQ with a sampling rate of 200 per second.

For the present analysis, ground motion records of EQs happened between 2004 and 2017, available on PESMOS are used. For estimating site characteristics, 341 records from 86 EQs, with magnitudes ranging from Mw=2.3 to 5.8, having focal depths ranging from 2 to 80km are used. Further, these records are corresponding to 101 recording stations, located in the hypocentral distance ranging from 9 to 355km. Coordinates of each of the recording station, used in this work are listed in Table 1, columns 2 and 3. Further, details of EQs used for estimating site characteristics are summarized in Table 2.

For estimating path attenuation however, only those EQs recorded at atleast two recording stations among which at least one recording station is located within hypocentral distance equal to or less than the reference distance (reference distance is discussed under section 'Spectral attenuation with distance) are considered. Out of 341 EQ records used for estimating site parameters, only 207 EQ records satisfies the above mentioned reference distance criteria. Final database for estimating path attenuation consists of 207 records from 32 EQs, recorded at 69 recording stations, with magnitude (Mw) in the range of 3.1 to 5.5, focal depths from 3 to 55km and hypocentral distance from 9 to 200km. Table 3 summarizes the details of the dataset used for estimating path attenuation. In addition, Figure 1 shows the source-to-recording station distance of the data set used in the present study.

**3.1 Data processing**

Signal to pre-event noise (all of equal window length) ratio (SNR) for all the records are computed and records with SNR≥5 (similar to Ameri et al., 2011) are considered for further analyses. All the EQ records are corrected for baseline correction following a 5% cosine taper and a band-pass filtering, between the frequency range of 0.25Hz and 15Hz, using a Butterworth filter. Further, time windows starting about 0.5s before the onset of the S wave and ending when 90% of the total seismic energy of the EQ record is reached, are separated and tapered with a 5% cosine window (Ameri et al., 2011; Bindi et al., 2009). Typical lengths of the time windows for the present analysis vary from 4 to 15s. Further, for some of the records, where the window lengths obtained to be longer than 15s, are fixed to 15s in order to minimize coda wave energy in the analysing time window (Oth et
al., 2008). Later, based on the extracted windows, the Fourier amplitude spectra is calculated for each EQ
record, smoothened by applying the Konno and Ohmachi (1999) algorithm, with the smoothing parameter "b" =
20.

For further analyses, path and site parameters are estimated using above processed EQ records, based on two
separate inversion procedures as discussed separately in the following sections.

**4 Path attenuation**

In the first step of inversion, path attenuation curves are developed by eliminating the effect of site parameter,
thereby retaining only the source and path attenuation characteristics. All EQ records, irrespective of whether
located on soil or rock sites can be utilized for inversion. This way, present method is very much suitable for
PESMOS database where accurate subsurface information of recording stations is not available. The horizontal
portion of the accelerograms (obtained by the root mean square average of the east-west and north-south
components) is considered for developing path attenuation curves. Detailed discussions on the method can be
found in following sub-section.

**4.1 Methodology**

Following Castro et al., (1990), observed spectral amplitude (acceleration) $U_{ij}(f, R_{ij})$, of EQ $j$, at recording
station $i$, and frequency $f$ can be modelled linearly as:

$$lnU_{ij}(f, R_{ij}) = lnM_i(f) + lnA(f, R_{ij}) \tag{1}$$

Here, $M_i(f)$ is a scalar, which is governed by the magnitude of the EQ (one value for each EQ). $A(f, R_{ij})$ is the
attenuation function and is independent of the magnitude of EQ. Here, $A(f, R_{ij})$ incorporates both geometric
spreading and anelastic attenuation variation with the hypocentral distance. It has to be mentioned here that
$A(f, R_{ij})$ in Eq. (1) is not limited to a particular functional form, instead, is assumed to decay smoothly with
hypocentral distance ($R_{ij}$) and thus take the value of unity at a reference distance ($R_0$), as given in Eq. 2 (Castro
et al., 1990; 1996; 2003).

$$A(f, R_0) = 1 \tag{2}$$

Model given by Eq. (1) has no factor representing site effect and is contained in both $A(f, R_{ij})$
and $M_i(f)$. Any rapid undulations in $A(f, R_{ij})$ are due to the absorbed site effects (Oth et al., 2008). Two
weighing factors, $w_1$ and $w_2$ are incorporated in the Eq. (1) following Castro et al., (1990). $w_2$ is used to
smoothen the attenuation term with distance curve by supressing the undulations and thereby removing any
absorbed site effects from $A(f, R_{ij})$. $w_2$ is used to impose $A(f, R_0) = 1$ constraint, as mentioned earlier. The
value of $w_1$ and $w_2$ here is chosen reasonably such that the site effects are supressed but the change in the
attenuation characteristics with distance can be observed (Oth et al., 2008). Solution to Eq. (1) in the matrix
form, after incorporating weighing factors, is obtained using singular value decomposition method, (discussed in
detail in Appendix A).

**4.2 Spectral attenuation with distance**

Figure 2 shows the number of EQ records for various hypocentral distance range considered. It can be observed
from Figure 2 that there are very less EQ records available beyond hypocentral distance of 115km. For this
reason, EQ records with hypocentral distance up to 115km are only considered for the determination of path
attenuation. The constraint $A(f, R_0) = 1$ is applied at $R_0$=15km, irrespective of the frequency. The hypocentral
distance range from 15 to 115km is divided into 10 bins, each bin having 10km width. Further, attenuation
curves are computed for each of the selected 17 frequencies from 1Hz to 15Hz (see Table 4, column 1).
Variation of attenuation curves with hypocentral distance, obtained in the present study, for the selected
frequencies is depicted in Figure 3. Based on Figure 3, a general trend in which attenuation curves exhibit decay
with hypocentral distance up to 105km can be observed. Beyond 105km a kink is observed as seen in Figure 3.
This kink in the attenuation curves beyond 105km is very distinct and clear at lower frequencies (<5.5 Hz).
Bindi et al., (2004) and Oth et al., (2011) reported a similar trend in the attenuation curves for the Umbria
Marche and Japan regions respectively. Oth et al., (2011) attributed this behaviour to the combined effect of
reflected or refracted wave arrivals from the Moho in Japan. Presence of Moho in the North-west Himalaya was
reported by Saikia et al., (2016) based on Teleseismic receiver function analysis. Referring to Oth et al., (2011)
work, presence of kink beyond 105km in attenuation curves, as obtained in this study may also be due reflected
or refracted waves from the Moho. Further, detailed study in this direction can be done in the future and is
beyond the presence scope of the work. Observing the attenuation curves at different frequencies as given in
Figure 3, it can concluded that attenuation curves at higher frequencies (>5.5Hz) decay more rapidly compared
to lower frequencies for the present study region. This observation is consistent with the findings by Castro et
al., (2003) for Guadeloupe (France) and Oth et al., (2011) for Japan.

[revised manuscript text omitted]

For $n^{th}$ earthquake

$$
\left|
\begin{array}{ccccccccccccc}
 & & & & & & & & & & & \vdots & \\
\end{array}
\right.
$$

| For $n^{th}$ earthquake | | | | | | | | | | | | | $s_n(f_n)$ | | $d_n(f_m)$ |
|---|---|---|---|---|---|---|---|---|---|---|---|---|---|---|---|
| 0 | 0 | … | 0 | ….. | 1 | | … | 0 | 1 | 0 | … | 0 | $g(f_1)$ | | $d_n(f_m)$ |
| 0 | 0 | … | 0 | ….. | 0 | 1 | … | 0 | 0 | 1 | … | 0 | $g(f_2)$ | | $\vdots$ |
| $\vdots$ | $\vdots$ | | $\vdots$ | | $\vdots$ | $\vdots$ | | $\vdots$ | $\vdots$ | $\vdots$ | | $\vdots$ | $\vdots$ | | $\vdots$ |
| $\vdots$ | $\vdots$ | | $\vdots$ | | $\vdots$ | $\vdots$ | | $\vdots$ | $\vdots$ | $\vdots$ | | $\vdots$ | $\vdots$ | | $\vdots$ |
| 0 | 0 | … | 0 | … | 0 | 0 | … | 1 | 0 | 0 | … | 1 | $g(f_m)$ | | $d_n(f_m)$ |

(12)

The matrix form in Eq. (12) represents a purely under determinate system since there are $(n+1) \times m$ parameters for '$m \times n$' data (here $m$ is the number of sample frequency and $n$ is the number of EQs recorded at a particular recording station). In here, Eq. (12) is solved using Moore- Penrose matrix inversion procedure (minimum norm inversion) given by Penrose, (1955) to determine $g(f)_j$ at each of the recording station.

Based on the above discussed methodology, inversions are performed for east-west, north-south and vertical components of EQ records separately to obtain the amplification curves in the frequency range of 0.25Hz to 15Hz, for each of the three components. For further calculation, the horizontal component is obtained as the geometric mean of east-west and north-south components.

**5.2 HVSR**

HVSR method is an extension of Nakamura (1989) technique, which is widely to assess the subsoil characteristics using recorded ambient noises. Nakamura (1989) technique is based on the assumption that the soil amplification effects are retained only in the horizontal component whereas the source and the path effects are maintained both in vertical as well as horizontal components of ground motion. Hence, the ratio of horizontal and vertical components gives an estimate of site amplification. Lermo and Chavez-Garccia (1993) extended Nakamura (1989) technique to S wave part of the accelerograms and studied the theoretical basis of the technique by numerical modelling oSV waves. Later, HVSR method was applied to EQ recordings worldwide (Luzi et al., 2011; Yaghmaei-Sabegh and Tsang 2011; D'Alessandro et al., 2012; Harinarayan and Kumar 2017a, b, 2018 etc.) to obtain the site characteristics.

Comparative studies between HVSR and other methods of evaluating site parameters reported by Field and Jacob (1995), Parolai et al., (2004), Shoji and Kamiyama (2002) Harinarayan and Kumar (2017b) etc. show that, HVSR can provide good and reliable estimate of predominant frequency. However, the above literatures also point out discrepancies in amplification levels obtained from HVSR with other methods. In order to compare the site amplification functions obtained from HVSR and GINV methods, HVSR for each station is computed considering the same S wave window as used in the GINV method. HVSR for each recording station is determined using the following steps;

1.  Calculate the response spectra considering 5% damping, for all the three components (north-south, east-
west and vertical) of ground motion records.

2.  Obtain the geometric mean of the two horizontal response spectra components (H) using Eq. (13) given
below;

$$H = (H_{EW} \times H_{NS})^{0.5} \tag{13}$$

3.  Calculate the ratio of $H$ to $V$ ($H/V$).

Where, $H_{EW}$ and $H_{NS}$ are the response spectrum of the horizontal east-west and north-south components respectively and V is the response spectrum corresponding to vertical component of ground motion. Then, the

HVSR at each of the recording station can be estimated as;

[revised manuscript text omitted]
 characterization of EQ recording stations of PESMOS located in the north-west Himalayas and adjoining area is attempted. EQ recorded between 2004 and 2017 are used in the analyses based on two-step inversion. While determining path attenuation, a kink at 105km hypocentral distance is observed in this work. Referring to similar observations from other regions, presence of Moho discontinuity is proposed in the region. This finding can be validated based on detailed study and is beyond the scope of present work objective. Further, based on attenuation curve obtained till 105km hypocentral distance and over wide range of frequencies, $Q_s = (105 \pm 11) f^{(0.94 \pm 0.08)}$ is obtained for the present study area, clearly indicating that the region is heterogeneous and seismically active.

In absence of proper geological information of PESMOS recording stations as highlighted by numerous recent studies, site classification of all the recording stations considered in this study are done based on GINV and HVSR. Based on this analysis, out of 101 recording stations, 10 are found to be located on rock while 91 stations are found to be located on soil sites. 1:1 matching for all the recording station from GINV and HVSR further enhances the confidence on present findings. Based on the findings, two empirical correlations for $A_{peak}$ and $f_{peak}$ with $V_{s30}$, are proposed.

Path attenuation and site characteristics are the key factors to be used for developing regional ground motion model. Further, with EQ catalogue known, present findings can be used for detailed seismic hazard assessment of the study area. In addition, identifying recording station whether considered recording stations are located on rock sites or soil sites will help in utilizing ground motion records for scenario based seismic hazard assessment.

**Authors Contribution:**

Harinarayan N H developed code generalized inversion, analyzed the records and all relevant literature review. Kumar Abhishek (AK) highlighted the importance of site characterization for PESMOS recording stations and need for the study.

**Acknowledgement**

The authors would like to thank the INSPIRE Faculty program by the Department of Science and Technology (DST), Government of India for the funding project ''Propagation path characterization and determination of in-situ slips along different active faults in the Shillong Plateau'' ref. no. DST/INSPIRE/04/2014/002617 [IFA14-ENG-104] for providing necessary funding and motivation for the present study.

**Appendix A**

In the matrix form, following the notations of Menke (1989) and incorporating the weighting factors
$w_1$ and $w_2$, Eq. (1) can be written in accordance with Castro et al., (1990) as:

 (A)                          (X)       (b)

$$
\begin{bmatrix}
& 0 & 0 & . & \dots & 1 & 0 & 0 & . & \dots \\
& 1 & 0 & . & \dots & 1 & 0 & 0 & . & \dots \\
. & . & . & . & \dots & . & . & . & . & \dots \\
& 0 & 0 & . & \dots & 0 & 1 & 0 & . & \dots \\
. & . & . & . & . & 0 & 1 & 0 & . & \dots \\
. & . & . & . & . & . & . & . & . & . \\
. & . & . & . & . & . & . & . & . \\
w_1 & 0 & 0 & . & \dots & . & . & . & . & \dots \\
-w_2/2 & w_2 & -w_2/2 & . & \dots & . & . & . & . & \dots \\
& -w_2/2 & w_2 & -w_2/2 & \dots & . & . & . & . & \dots \\
. & . & . & . & \dots & . & . & . & . & \dots
\end{bmatrix}
\begin{bmatrix}
\ln A_1 \\
. \\
. \\
. \\
\ln A_{10} \\
\\
\ln M_1 \\
. \\
. \\
. \\
\ln M_N
\end{bmatrix}
=
\begin{bmatrix}
\ln U_{11} \\
. \\
. \\
\ln U_{ij} \\
. \\
\\
\\
\\
\\
\\
.
\end{bmatrix}
$$

                                                  (A1)

The hypocentral distances of the data set is discretized into number of bins of equal widths and the value of
$A(f, R_{ij})$ is computed at each bin. The width of the bins are selected such that there is almost equal number of
data points in every bin. Further, $\ln A(f, R_{ij})$ versus hypocentral distance curves at each of the selected
frequencies are computed solving Eq. (2) in a least square sense, using singular value decomposition method
(Menke, 1989).

**FIGURES**

[Figure]

Figure 1: Map of the region under study with EQs (stars), recording stations (triangles), and paths (solid-lines).

[Figure]

**Figure 2: Distribution of hypocentral distances in the data set.**

[Figure]

**Figure 3: S wave spectral attenuation versus hypocentral distance. Note that ln A(f,R₀) at reference distance is zero.**

[Figure]

**Figure 4: Frequency dependence of the quality factor Q for hypocentral distances between 15 km to 105 km**

[Figure]

**Figure 5:  Comparison of $Q_S$ values of North West Himalaya with those obtained from parts of North West Himalaya**
**and Delhi region. The compared relations for Qs versus frequency are as follows: Garhwal-Kumouan Himalaya:**
**$Q_S = 175 * f^{0.833}$ (Mukhopadhyay et al., 2010); Kinnaur Himalaya: $Q_S = 86 * f^{0.96}$ (Kumar et al., 2014) ; Garhwal**
**Himalaya: $Q_S = 151 * f^{0.84}$ (Negi et al., 2015); Delhi and NCR region: $Q_S = 98 * f^{1.07}$ (Sharma et al., 2015).**

[Figure]

**Figure 6: Comparison of Q$_S$ values of this study with regions of different tectonic settings of the world.**

[Figure]

[Figure]

[Figure]

[Figure]

[Figure]

**Figure 7 (a-f). Site amplitude curves obtained using GINV for horizontal component and vertical component**

[revised manuscript text omitted]

           **Pandey et al., 2016a; * Pandey et al., 2016b

**List of Figures**

**List of Tables**